



**Measurement report: Seasonal, distribution and sources of**
**organophosphate esters in PM$_{2.5}$ from an inland urban city in**
**southwest China**
Hongling Yin, Jiangfeng Liang, Di Wu, Shiping Li, Yi Luo, Xu Deng
College of Resources and Environment, Chengdu University of Information Technology, Chengdu,
Sichuan, 610025, China
*Correspondence: Hongling Yin (yhl@cuit.edu.cn)*
**Abstract.** Organophosphate esters (OPEs) are emerging contaminants in recent years and studies
concluded that urban centers were a significant source of OPEs. Samples were collected from six
ground-based sites located in Chengdu, a typical fast developing metropolitan of southwest China and
were analyzed for seven OPEs in atmospheric PM$_{2.5}$. The concentrations of $\Sigma_7$OPEs in PM$_{2.5}$ ranged
from 5.83 to 6.91 ng·m$^{-3}$, with a mean of 6.6 ± 3.3 ng·m$^{-3}$, and the primary pollutants were TBEP,
TnBP, TCEP and TCPP which made up more than 80% in the $\Sigma_7$OPEs. The concentrations of $\Sigma_7$OPEs
were higher in autumn/winter than that in summer. Nonparametric test showed that there was no
significant difference in $\Sigma_7$OPEs concentrations among the six sampling sites, but the occurence of
unexpected high level of individual OPEs at different sites in autumn might indicate that there was a
noteworthy emission. Very strong correlation ($R^2 = 0.98$, $p<0.01$) between the OPEs in soil and in
PM$_{2.5}$ suggested the atmospheric PM$_{2.5}$ settlement is an important source of OPEs in soil. The
backward trajectory analysis displayed that OPEs in PM$_{2.5}$ were mainly affected by local sources. The
principal component analysis (PCA) identified the OPEs in PM$_{2.5}$ were largely sourced from the plastic
industry/interior decoration /traffic emission (34.5%) and chemical, mechanical and electrical industry
(27.8%), while PMF model found the main sources were the plastics industry/indoor source emissions,
the food/cosmetics industry, and industrial emissions. Differed from the coastal cities, the sustained
and stable high local emissions in the inland city were identified which were particularly noteworthy.
The chlorinated phosphate, especially TCPP and TCEP have a high content, whose usage and source
emissions should be controlled.



## 1. Introduction

With the prohibition of brominated flame retardants, the production and the demand of organophosphate esters (OPEs) have rapidly increased in recent years (Wang et al., 2012). To date, OPEs are widely distributed in the environment and have been detected in air (Bacaloni, A. et al., 2008), water (Wang et al., 2013; Li et al., 2014), soil (Yin et al., 2016), sediment (Cristale J. et al., 2013; Celano R, et al., 2014) and organisms (Araki et al., 2014; Kim et al., 2011). However, many scholars found that OPEs have negative effects on the human body with the characteristics of water resistance, weather resistance, heat resistance and good polymer substrates compatibility (Matthews, et al., 1990; 1993). Some countries have legislated to restrict the usage of OPEs. Nevertheless, the production and usage of OPEs in China is still on the rise.

As synthetic substances, the only source of OPEs in the environment is anthropogenic emissions. The detection of OPEs in Arctic and Antarctic snow samples and atmospheric particulate matter samples demonstrated that OPEs can be transported over long distances. Studies on OPEs in oceans were carried out a lot, and the concentrations of particle-bound OPEs ranged from tens to thousands of ng m$^{-3}$ (Covaci et al., 2007; Cristale J & Lacorte S., 2013; Li et al., 2017). Researchers noted that the contribution of air flow originated from the mainland when high concentrations of OPEs (thousands of ng m$^{-3}$) appeared (Möller et al., 2012; Lai et al., 2015). In addition, studies proved the urban area was the highest pollution area of OPEs. However, until now, only a few papers reported the concentration and distribution of OPEs in urban atmospheric PM$_{2.5}$. Concentrations of OPEs in most cities were lower than 10 ng m$^{-3}$, higher concentrations of 19.2 ng m$^{-3}$ were observed at a suburban site in Shanghai, and 49.1 ng m$^{-3}$ were observed in Hongkong (Ohura et al., 2006; Salamova et al., 2014b; Marklund et al., 2005; Shoeib et al., 2014; Yin et al., 2015; Liu et al., 2016; Ren et al., 2016; Guo et al., 2016; Wong et al., 2018). To date, most of studies in China focus on the OPEs in the Yangtze River Delta and Pearl River Delta, especially eastern coastal cities while little attention was paid to the western inland cities.

Chengdu is a typical inland city located in the southwest of China. It is the capital and megacity of Sichuan Province, which covers an area of 14335 square kilometers and has a permanent population of 16.33 million. As the important national high-tech industrial base, commercial logistics center and comprehensive transportation hub determined by the State Council, Chengdu is the important central



city in the western region. Liu et al. (2016) reported an investigation of three chlorinated OPEs in the
atmosphere at 10 urban sites in China during 2013–2014 and found that the highest annual mean
concentrations were observed in Chengdu ($1,300 \pm 2,800$ ng m$^{-3}$). However, there is still a lack of
information regarding the levels, sources, and fate of OPEs in the southwest China which may
obviously differed from the coastal cities or over the sea. In this study, we investigated the atmospheric
OPEs in PM$_{2.5}$ through intensive sampling in an economically fast growing city--Chengdu. Sampling
was carried out over one year (October 2014 to September 2015) which was a continuous and further
project of our previous study from December 2013 to October 2014. The aims of the study were to: a)
report the levels and composition profiles of OPEs in urban air in the typical inland city; (b) obtain the
seasonal and spatial variation of OPEs in PM$_{2.5}$; (c) investigate the relationships and correlations
among the target compounds or with influence factors; (d) illustrate the potential sources of OPEs in
PM$_{2.5}$.

**2. Materials and Methods**

**2.1 Chemicals**

The main reagents, such as ethyl acetate, acetone, hexane and acetonitrile, were High Performance
Liquid Chromatography (HPLC) grade (Kelon Chemical, China). The standard solution (Sigma
Aldrich) included tri-n-butyl phosphate (TnBP), tris-(2-ethylhexyl)phosphate (TEHP), tris-(2-
butoxyethyl) phosphate (TBEP), triphenyl phosphate (TPhP), tris-(2-chloroethyl)-phosphate (TCEP),
tris-(2-chloroisopropyl)phosphate (TCPP), and tris-(2.3-dichloropropyl)-phosphate (TDCIPP). Copper,
aluminium oxide, silica gel, Na$_2$SO$_4$ and other chemicals were purchased from Kelon Chemical.
Deionized water was supplied from a Milli-Q equipment.

**2.2. Sample collection**

The atmospheric sampling sites were located in the main city area (site B: downtown; site C: south; site
D: east; site E: north; site F: west) and suburban area (site A) of Chengdu, as shown in Fig. S1. The
atmospheric samples were collected by KC-6120 medium flow atmospheric comprehensive sampler
with quartz film. The speed was set at 100 L min$^{-1}$, and each collection campaign lasted 23 h. The
sampling campaign was carried out between October 2014 and September 2015. A total of 149
samples were obtained. Most of the weather conditions were cloudy days, with south/north wind at



≤5.5 m/s. Temperature ranged from 0 to 35 ℃. Weather conditions could represent typical weather
conditions of the season.

## 2.3. Sample preparation and analysis

The shredded $PM_{2.5}$ sample film was placed in a test tube and incubated in 20 mL ethyl acetate/acetone
(v:v, 3:2) for 12 hours. After ultrasonic extraction for 30 minutes, the liquid was separated, and the
residue was further extracted with 10 mL ethyl acetate/acetone (v:v, 3:2) by ultrasonic extraction for 15
minutes. The extracts were combined and concentrated by vacuum-condensing equipment (Buchi
Syncore Q-101, Switzerland) to approximately 1 mL, then loaded onto an activated aluminium
oxide/silica gel (v: v, 3: 1) column. The column was first eluted with 20 mL hexane to remove
impurities, then with 20 mL ethyl acetate/acetone (v: v, 3: 2) and the eluate (ethyl acetate/acetone) was
collected. The solvent extracts were concentrated by vacuum-condensing equipment and diluted to 200
$\mu$L for gas chromatography-mass spectrometry (GC-MS) (Shimadzu 2010plus, Japan) analysis.
The GC is equipped with a capillary column Rti-5MS (30 m × 0.25 μm × 0.25 mm, Kelong), with a
280 ℃ inlet temperature using splitless injection. The MS source was electron impact (EI) and
operated in selected ion monitoring (SIM) mode. Helium was used as a carrier gas with a flow rate of
1.00 mL $min^{-1}$. The GC oven temperature was held at 50 ℃ for 1 minute, increased to 200 ℃ at 15 ℃
$min^{-1}$ and held for 1 minute, increased to 250 ℃ at 4.00 ℃ $min^{-1}$, and then increased to 300 ℃ at 20 ℃
$min^{-1}$ and held for 4 minutes. The interface temperature was 280 ℃, and the ion source temperature was
200 ℃. The respective characteristic ion and reference ions (m/z) of the 7 target compounds were:
155/99, 211, 125 (TnBP), 249/63, 143, 251 (TCEP), 125/99, 201, 277, 157 (TCPP), 75/99, 191, 209,
381 (TDCPP), 326/325, 77, 215 (TPhP), 85/100, 199, 299 (TBEP), 99/113 and 211 (TEHP).

## 2.4. QA / QC

The concentrations of the 7 OPEs were determined by an external standard method. The correlation
coefficients of the standard curves of the seven OPE monomers were all greater than 0.990. The
recoveries of the 7 OPEs ranged from 83.9% to 121.2%. A matrix blank was analysed with each batch
of samples. Only TnBP was detected in the blanks, and the level of TnBP found in the blanks was <5%
of the concentrations measured in all samples, which means it was negligible. The instrument precision
was in the range of 1.9%-8.3%.



## 3. Results and Discussion

### 3.1. Levels of OPEs in PM$_{2.5}$

OPEs were present in PM$_{2.5}$ samples collected across the study area (Fig. S1). Four OPEs (TCPP, TDCPP, TCEP and TnBP) were detected in all samples (n=149), while TBEP was detected in all but one sample. Additionally, TEHP was detected in 96.7% of samples overall and TPhP was detected in 98% of samples. The high detection frequencies of most OPEs indicated OPE contamination was ubiquitous in the air of Chengdu city.

Concentrations of Σ7OPEs in PM$_{2.5}$ across the six sites were in the range of 3.5 - 11.5 ng m$^{-3}$, and the annual median concentration of Σ$_7$OPEs was 6.5 ± 3.3 ng m$^{-3}$ (Fig. 1). The average value of OPEs in PM$_{2.5}$ at each site in four seasons was almost at the same level (5.8 ± 1.3 ng m$^{-3}$-6.9 ± 2.5 ng m$^{-3}$). Nonparametric test showed that there was no significant difference in Σ$_7$OPEs concentrations among the six sampling sites, indicating that the atmosphere mixed evenly, and there was no particularly heavy or light pollution area in Chengdu city. These data are quite consistent with our previous study that showed the annual median concentration of OPEs in PM$_{2.5}$ from December 2013 to October 2014 (Yin et al., 2015). Interestingly, the annual median of total OPEs at the suburban site was not the lowest as might be expected and is instead likewise similar to, or even higher than some urban sites which indicated more local sources of these compounds in the suburban area.

The concentrations of OPEs in the particles of Chengdu were comparable to that reported from Beijing (0.257 - 8.36 ng m$^{-3}$) (Wang et al., 2018), 6.6 ng m$^{-3}$ (Σ$_6$ OPEs) for Shanghai urban site (Ren et al., 2016), 6.5 ng m$^{-3}$ (Σ$_6$ OPEs) for Bursa, but higher than that in Houston, US (Σ$_{12}$ OPEs, 0.16 - 2.4 ng m$^{-3}$) (Clark et al., 2017), Dalian (Σ$_9$ OPEs, 0.32-3.46 ng m$^{-3}$,1.21 ± 0.67 ng m$^{-3}$) (Wang et al., 2019), European Arctic(0.033 - 1.45 ng m$^{-3}$) (Salamova et al., 2014), Northern Pacific and Indian Ocean (0.23 - 2.9 ng m$^{-3}$) (Moller et al., 2012), the Yellow Sea and Bohai Sea (0.044 - 0.52 ng m$^{-3}$) (Li et al., 2017), South China Sea (0.047 - 0.161 ng m$^{-3}$) (Lai et al., 2015), North Atlantic and Arctic Oceans(0.035 - 0.343 ng m$^{-3}$) (Li et al., 2017). And lower than that in Guangzhou and Taiyuan (Σ$_{11}$OPEs, 3.10 - 544ng m$^{-3}$) (Chen et al., 2020), in Bursa, Turkey (Σ$_6$OPEs,0.53 - 19.14 ng m$^{-3}$) (Kurtkarakus et al., 2018), 20 industrial sites in an urban region (Σ$_{12}$OPEs, 0.52 - 62.75 ng m$^{-3}$) in Guangzhou, China(Wang, T. et al., 2018).



## 3.2. The composition profiles of OPEs in PM$_{2.5}$

There was clear dominant non-chlorinated OPEs across Chengdu city. The annual median concentrations of total OPEs were fairly uniform at six sites and influenced mainly by the alkylated OPEs. As listed in Table 1, the general trend was found that TBEP was the most abundant OPE (2.3 ng m$^{-3}$, 35.3%), followed by TCEP (1.1 ng m$^{-3}$, 16.3%) ≈ TnBP (1.0 ng m$^{-3}$, 15.6%) ≈ TCPP (1.0 ng m$^{-3}$, 15.0%) > TPhP (0.5 ng m$^{-3}$, 8.4%) > TEHP (0.3 ng m$^{-3}$, 5.1%) > TDCPP (0.3 ng m$^{-3}$, 4.3%), with the concentrations of TBEP being approximately 7 - 10 times higher than those of TDCPP and TEHP. The composition profile of OPEs was similar at all sites except for that the east site which has a higher contribution of TnBP. But TBEP, TCEP, TCPP and TnBP were dominant OPEs across the city who contributed more than 80% to Σ$_7$OPEs. This profile was similar to that in Longyearbyen, Norway, with primary pollutants being TnBP and TBEP (Möller et al., 2012), as well as the OPEs in outdoor urban air being TBEP > TCPP > TCEP > TnBP > TPhP in Stockholm, Sweden (Wong et al., 2018) and TBEP > TCPP > TPhP > TEHP > TCEP in Turkey (Kurtkarakus et al., 2018). However, these results substantially differed from the report of an urban site in Shanghai that showed TCEP (0.1 - 10.1 ng m$^{-3}$, 1.8 ng m$^{-3}$) > TCPP (0.1 - 9.7 ng m$^{-3}$, 1.0 ng m$^{-3}$) > TPhP (0.06 - 14.0 ng m$^{-3}$, 0.5 ng m$^{-3}$) > TBP (0.06 - 2.1 ng m$^{-3}$, 0.4 ng m$^{-3}$) > TDCPP (Nd.-23.9 ng m$^{-3}$, 0.3 ng m$^{-3}$), whereas TBEP was only detected in 3 out of 116 samples (Nd.-0.7 ng m$^{-3}$, Nd.) (Ren et al., 2016), and the reported data over the Bohai and Yellow Seas showed TCPP (43 - 530 ng m$^{-3}$; 100 ng m$^{-3}$, 50 ± 11%)> TCEP (27 - 150 ng m$^{-3}$; 71 ng m$^{-3}$, 25 ± 7%) > TiBP (19 - 210 ng m$^{-3}$; 57 ng m$^{-3}$, 14 ± 12%) > TnBP (3.0-37 ng m$^{-3}$; 13 ng m$^{-3}$). Li et al. (2014) determined the primary pollutant of outdoor air in Nanjing was TCEP, and TBEP was not detected. These differences reflected that there were significant differences in OPE production and usage in different regions, even in the same country. It should be noted that concentrations of TCPP and TCEP were in the same level in this study, suggesting the industrial replacement of TCEP by TCPP wasn't identified in the southwest China which differed from that the higher concentration of TCPP in comparison with TCEP was observed due to the industrial replacement of TCEP by TCPP in Europe (Quednow and Püttmann, 2009). This was confirmed by the fact that there are manufacturers and sellers of TCEP and TCPP in Chengdu, indicating that there is production and demand both for TCPP and TCEP in this region.



Combined with the data of 2013-2014 year (Yin et al., 2015), TBEP was always the dominant OPEs
during the two sampling periods (2013-2014 and 2014-2015). Kruskal Wallis test was used and found
that TnBP and TCPP had no significant difference between the two sampling periods, but there were
significant differences in other kinds of OPEs between the two sampling periods. This indicated that
the production and usage of individual OPEs have certain change suggesting that OPEs should be
better investigated and governed for individual compounds.
OPEs can be categorized by whether they are halogenated, alkylated or aryl OPEs. Of the OPEs
measured in this study, TCEP, TCPP and TDCPP are halogenated, TBEP, TnBP and TEHP are
alkylated, and TPhP is aryl OPEs. The OPEs in $PM_{2.5}$ at all sites were dominated by the alkylated
compounds ($55.9 \pm 10.1\%$), followed by halogenated OPEs ($35.8 \pm 9.9\%$) and aryl OPEs ($8.3 \pm 4.1\%$).
Our results are similar to those observed in Bursa, Turkey (Kurtkarakus et al., 2018), whose alkylated
OPEs covered $68\% \sim 95\%$ of total OPEs, while halogenated OPEs covered $3.1\% \sim 29\%$, and aryl OPEs
covered $1.4\% \sim 3.7\%$ of total OPEs. At Longyearbyen, the non-chlorinated OPE concentrations
comprised 75% of the $\Sigma_8$OPE concentrations (Salamova et al., 2014a). However, our results are
obviously different from many studies with the atmospheric samples collected in urban areas being
dominated by chlorinated OPEs ($50 \sim 80\%$) (Salamova et al., 2014b; Liu et al., 2016; Guo et al., 2016).
In our study, non-chlorinated OPEs were dominant in urban and suburban area across the city.
**3.3. Seasonal and spatial variation of OPEs in $PM_{2.5}$**
The mean seasonal concentrations were plotted for six sampling sites in Fig. 2. The data were quite
consistent with our previous study from December 2013 to October 2014 (Yin et al., 2015). The
concentrations of OPEs in $PM_{2.5}$ have been fairly uniform in the past three years. As shown in Fig. 2,
the general order of the decreasing average $\Sigma_7$OPEs concentrations in suburban area was autumn ($8.4 \pm$
$4.3$ ng m$^{-3}$) $\approx$ winter ($8.4 \pm 4.5$ ng m$^{-3}$) > spring ($7.6 \pm 2.2$ ng m$^{-3}$) > summer ($3.5 \pm 1.1$ ng m$^{-3}$), while in
urban area was autumn ($9.30 \pm 3.89$ ng m$^{-3}$) > winter ($6.63 \pm 3.65$ ng m$^{-3}$) > spring ($6.36 \pm 1.72$ ng m$^{-3}$)
> summer ($4.60 \pm 1.91$ ng m$^{-3}$). The average concentration of $\Sigma7$OPEs in autumn/winter was
approximately 2 times that in summer. In summer, the turbulent flow accelerated the diffusion of
pollutants, leading to the lowest concentration, while the higher concentrations of OPEs appeared in
autumn and winter because the inversion layer appeared more frequently in autumn and winter,
resulting in the pollutants being more difficult to diffuse and dilute. This seasonal variation was mostly



in line with that at the Shanghai urban site of autumn (8.4 ng m$^{-3}$) > winter (7.6 ng m$^{-3}$) > spring (5.5 ng
m$^{-3}$) > summer (4.4 ng m$^{-3}$), of which the maximum value was also approximately twice the minimum
(Ren et al., 2016). In addition, this finding was similar to that in Xinxiang that no significant seasonal
changes and only exhibited individual high values in winter. On the contrary, Wang et al. (2019) found
the PM$_{2.5}$-bound fractions of OPEs varied significantly between seasons in Dalian, China, with their
concentrations higher in hot seasons, which may due to the temperature-driven emission or gas-particle
partitioning. Wong et al. (2018) reported that most of OPEs in outdoor urban air showed seasonality,
with increased concentrations during the warm period in Stockholm, Sweden. Sühring et al. (2016)
reported temperature dependence of chlorinated OPEs and EHDPP in Arctic air. Liu et al. (2014) did
not observe any temperature dependence for the OPEs in urban air in Toronto, Canada. Thus previous
reports of temperature dependence of OPEs are not consistent. In this study, the lowest concentrations
of Σ$_7$OPEs and individual compound were observed in summer suggesting the OPEs level was not
driven by the temperature-driven emission or gas-particle partitioning, but mainly by the local emission
sources.
Compared to the coastal cities, the most obvious difference was that concentrations of almost all OPEs
monomers in this study were highest in autumn/winter and lowest and concentrated in summer
suggesting the sustained and stable high local emissions in the inland city which were particularly
noteworthy. No point source was identified in summer and the OPEs level was diluted and diffused in
summer due to the higher wind speed than in winter in the inland city. This was different from the
coastal cities: Liu et al. (2016) reported that the highest TCPP and TCEP concentrations were observed
in the summer in Guangzhou and Javier et al. (2018) found the OPEs in spring generally exhibited the
lowest concentrations in Bizerte, Tunisia, probably linked to the influence of local meteorological
conditions and air mass trajectories to a lesser extent.
Though Kruskal Wallis test showed that there was no significant variation of Σ$_7$OPEs concentrations
across the city, the spatial differences were identified in the study. For example, TnBP and TCPP had
significant difference among six sites. In addition, the higher concentrations and more dispersed pattern
of most OPEs were observed in autumn and winter than in summer (Fig. 3). The concentrations of
TEHP in autumn at the eastern and northern sampling site were more dispersed than others. The same
dispersion pattern was observed for TBEP in winter at the western sampling site, TPhP in autumn at
the suburban sampling site, TnBP in autumn at the eastern sampling site, suggesting that there existed



the extra emission sources in autumn or winter. Considered the layout of Chengdu which develops
from the central area with the loop line (the first ring road, the second ring road and the third Ring
Road), we could understand the OPEs levels and distribution were quite uniform across the city. But
different types of industrial parks in different directions in Chengdu may be the reason for the spatial
differences of OPEs. For example, in the east of Chengdu, there are automobile industrial parks and
other large industrial parks while logistics and shoemaking industrial parks in the suburbs. The
occurance of unexpected high level of individual OPEs at different sites in autumn might indicate that
there was a noteworthy emission. The spatial and seasonal variation of individual OPE suggest that the
control and management of OPEs should be taken to the individual OPE.
OPEs can be categorized as halogenated, alkylated and aryl OPEs. Of the OPEs measured in this study,
TCEP, TCPP and TDCPP are halogenated, TBEP, TnBP and TEHP are alkylated, and TPhP are aryl
OPEs. Many studies focused on the halogenated OPEs due to their persistence, bio-accumulation, and
potential human health effects, and they dominated the OPEs profile in the air of many cities and other
areas (Zhang et al., 2016, Li et al., 2017). Liu et al. (2016) reported that the sum of the concentrations
of the three halogenated OPEs at 10 urban sites ranged from 0.05 to 12 ng m$^{-3}$ suggesting the highest
production volume and widest applications of OPEs leading to large emissions of OPEs in China in
recent years. However, in our study, the mean concentrations of halogenated, alkylated and aryl OPEs
were 2.4 ± 1.4 ng m$^{-3}$, 3.7 ± 2.1 ng m$^{-3}$, 0.5 ± 0.4 ng m$^{-3}$, respectively, which showed the alkylated
OPEs dominated the profile of OPEs in PM$_{2.5}$ in Chengdu. In different seasons, the most notable
seasonal variation was observed for alkyl phosphate, followed by halogenated OPEs and aryl OPEs.
These results were significantly different from those in other studies which reported that the
halogenated OPEs had the maximum seasonal variability (Guo et al., 2016; Shoeib et al., 2014).
**3.4. Correlation analysis of OPEs**
**3.4.1 Linkage to environmental factors**
Most of OPE monomers concentrations in PM$_{2.5}$ have a strong linear correlation (R$^2$ = 0.79) with their
vapor pressure (Fig. 4), suggesting that the vapor pressure is an important factor controlling the levels
of OPEs in PM$_{2.5}$ except for TBEP. Generally speaking, the greater the vapor pressure of OPEs, the
easier it is to be released into the environment. Therefore, the sources of most OPEs in Chengdu
atmospheric PM$_{2.5}$ are mainly both from the production process containing OPEs and the phase





transition process before they enter into the atmosphere. The boiling points of OPEs are relatively high,
so they tend to be adsorbed in $PM_{2.5}$ after being released to the environment, and their gas-particle
distribution determines their concentration in $PM_{2.5}$. Interestingly, the vapor pressure of TBEP is lower
than other OPEs, but its concentration in $PM_{2.5}$ was higher which indicated that there were sustained
and stable high emission sources to keep its concentration at a high level which may include the traffic
emission source (Chen et al., 2020). Sühring et al. (2016) reported non-halogenated OPE
concentrations in Canadian Arctic air appeared to have diffuse sources or local sources close to the
land-based sampling stations.

**3.4.2 Correlation between target analytes**

Spearman's ranks correlation coefficients were used to investigate the potential emission sources for
OPEs by the relationship between individual OPE in $PM_{2.5}$ (Fig.5, Table 2). Fig. 4 showed no
statistically significant positive correlations between OPE monomers (r<0.50, $p$<0.01). However,
$\Sigma_7$OPEs concentrations were closely related to TBEP, TCEP and TnBP (r=0.53-0.61, $p$<0.01) which
further identified the OPEs levels were influenced mainly by the dominated OPEs compounds.
Comparatively, weak correlations between most of OPEs were observed in urban regions (Wang et al.,
2018) and Turkey (KurtKarakus et al., 2018). However, strong correlations between individual OPEs
were found in Guangzhou and Taiyuan (Chen et al., 2020).
Further analysis results were shown in Table 2. Only significant correlation between TCPP and TCEP
both at downtown (r=0.82, $p$<0.01) and suburban sites (r=0.85, $p$<0.01) were observed indicating the
high homology between these two compounds. So the inland city in China is still using a large number
of products containing chlorinated flame retardants, which was confirmed by our previous study of
house dust (Liu et al. 2017; Yin et al., 2019). At downtown site, another significant correlation existed
between TEHP and TCEP (r=0.50, $p$<0.01) while others have weak to moderate correlations (r<0.46,
$p$<0.01). The downtown area mainly focuses on the light industry and software development, and
TCPP, TCEP, TnBP, TBEP and TPhP are used in textile, leather, electronic products and other fields.
However, the correlation of each OPE monomer at site A (suburb) was stronger than that in the urban
area. TnBP and TCEP, TnBP and TDCPP, TCEP and TCPP, TCEP and TDCPP, TCEP and TBEP,
TCPP and TDCPP and TBEP were all extremely significant. This result indicated that the pollution in
the suburb was commixed and was influenced by many kinds of pollution sources.





### 3.4.3 Correlation analysis of OPEs and $PM_{2.5}$ concentrations


The SPSS software scatter diagram was used to analyse the relationship between the concentrations of
OPE monomers and $PM_{2.5}$. As displayed in Fig.S2, only weak to moderate correlation were observed
between most of OPEs and $PM_{2.5}$ except significant correlation was found between TDCPP and $PM_{2.5}$
(r=0.53, $p$<0.01) which suggest the continuous and relative constant local sources were the main
sources. This result was similar with that reported from Taiyuan (Guo et al., 2016), where no
correlation was between the concentrations of OPEs and the concentration of particulate matter.
However, this result differed from that in Xinxiang (Shen et al., 2016), which showed that the
concentrations of OPEs and $PM_{2.5}$ had significant correlation (r=0.85c), and a high value of
OPEs/$PM_{2.5}$ was related to the contribution of the air mass from the heavily polluted area (Henan and
Jiangsu province), while low OPEs/$PM_{2.5}$ was due to the air mass from Shanxi-Gansu and Neimenggu
Province. Chen et al. (2020) found there was a significant correlation (p < 0.05) between the
concentrations of $\Sigma_{11}$OPEs and $PM_{2.5}$ in some sampling sites but not a site located in the urban region
in Guangzhou with potential additional pollution sources.

### 3.4.4 Correlation analysis of OPEs in $PM_{2.5}$ and soil


Due to the low detection frequency of TCPP and TDCPP in the soil (Yin et al., 2016), the relationship
of other five OPE monomers in the soil and in atmospheric $PM_{2.5}$ were presented in Fig. 6. A very
strong linear relationship was obtained between the OPEs in soil and in $PM_{2.5}$ ($R^2$ = 0.98, $p$<0.01 ),
indicating that the atmospheric $PM_{2.5}$ settlement is an important source of OPEs in the soil, so does the
soil be a source for OPEs in the air.

### 3.4.5 Correlation analysis of OPEs indoor and outdoor air


The OPEs profile in outdoor air in this study were: TBEP> TCEP > TnBP> TCPP > TPhP> TEHP>
TDCPP, which was different with indoor dust reported from our previous study (Liu et al., 2017):
TPhP>TCPP>TnBP> TDCPP >TBEP> TCEP > TEHP. TPhP is used as one of important alternatives
for technical decabrominated diphenyl ether (deca-BDE) product, which is typically used as a flame
retardant in electrical and electronic products. In addition, the use of plastic film and rubber may be an
important source of TPhP. Thus OPEs in indoor dust mainly comes from indoor environment and
related to human activities, not from outdoor air. Studies in Swedish (Wong, 2018) reported the



concentrations of OPEs in indoor air were TCPP > TCEP > TBEP > TnBP> TPhP, and in outdoor
urban air were TBEP > TCPP > TCEP > TnBP > TPhP (Wong, 2018) which also indicated the
differences of emission sources in indoor and outdoor air due to the different use of OPEs.
**3.5 Source apportionment of OPEs**
**3.5.1 Analysis of backward trajectory model**
The backward trajectory cluster analysis (HYSPLIT4) combines the horizontal and vertical motion of
the atmosphere, which can analyse the transport, migration and diffusion of atmospheric pollutants,
were used in this study. The height of AGL500m can better represent the characteristics of the process
wind field, and HYSPLIT4 was used to obtain the 24 hours backward trajectory of AGL500m during
the sampling period of Chengdu. During the sampling period, the air mass was mainly from the
northeastern and southern parts of Sichuan Province, including Mianyang, Deyang, Renshou and
Chengdu, and a few of the trajectories came from Chongqing and other places in Gansu Province.
Therefore, during the sampling period, Chengdu was mainly affected by the air mass of the eastern
Sichuan.
In different seasons, the air sources always came from the southern or the northern regions of Chengdu.
In spring, Chengdu was influenced by air mass from the southern region, which could be divided into
three paths: (a) from Ya'an through Renshou to Chengdu; (b) from Leshan and Yibin; and (c) from
Chongqing through Ziyang to Chengdu. The concentrations of OPEs at the northern and suburban site
were relatively high in spring. During the summer period, Chengdu was mainly influenced by air
masses from both the southern areas (Yibin, Zigong and others) and the northern areas (Gansu
Province, Guangyuan and Mianyang), but there was no significant difference in OPE concentrations at
each sampling site, nor in autumn and winter. Combined with the backward trajectory cluster analysis
and the concentrations of OPEs at each sampling site, the concentrations of OPEs had no obvious
change. This result suggested that OPEs were not affected by exogenous pollution but were mainly
affected by the local sources of Chengdu. These results are consistent with the meteorological and
topographic conditions. Chengdu's wind has always been breezy with much smaller strength than
coastal cities or other inland cities. The wind direction is relatively constant, mainly from the south and
the north. In addition, Chengdu is located in the basin, surrounded by the Qinghai-Tibet Plateau, the
Qinling Mountains, etc. These topographic and meteorological conditions block the influence of



foreign sources on Chengdu's atmosphere, which further explained that the pollution of OPEs in PM$_{2.5}$
was controlled by endogenous pollution, not by exogenous pollution.

**3.5.2 Principal Component Analysis**

The principal component analysis (PCA) of OPEs was carried out by SPSS. The normalized correlation
coefficient matrix of the original data of each sampling site showed that there was a strong correlation
between TCPP and TCEP, TCEP and TBEP, and TnBP and TPhP, which satisfied the condition of
dimensionality reduction of PCA. Two principal component factors were obtained in this study. The
cumulative contribution of the two principal component factors was 62.3%, which can basically reflect
the data information. The results were shown in Table S1. For factor 1, there was a large load on
TCEP、TCPP、TBEP and a moderate load on TDCPP. Factor 1 can represent the sources of OPEs
from the plastic industry, interior decoration and traffic emission, with the contribution ratio of 34.5%.
Factor 2 has higher load on TnBP, TEHP and TPhP. The highest load was on TnBP, which is often
used as a high-carbon alcohol defoamer, mostly in industries that do not come in contact with food and
cosmetics, as well as in antistatic agents and extractants of rare earth elements. TEHP can be used as an
antifoaming agent, hydraulic fluid and so on. TPhP is typically used in electrical and electronic
products, or plastic film and rubber. Factor 2 can be considered the chemical, mechanical and electrical
industry, and its contribution ratio was 27.8%.

**3.5.3 PMF model analysis**

The basic principle of the PMF method is to decompose the sample matrix into a factor contribution
matrix and factor component spectrum. The source type of the factor is judged according to the factor
component spectrum, and then the contribution ratio of source is determined. From 149 samples
collected in Chengdu, 132 valid samples were selected to participate in the model calculation and three
factors were determined. TPhP was the only chemical with residual (4.0) greater than 3. Concentrations
of OPEs satisfied the normal distribution. The components of factor 1 were complex. Factor 1
contributes 71.0%, 70.7% and 70.9% to TCEP, TCPP and TEHP, respectively, and 58.3% to TPhP.
Factor 1 was deduced to be the plastics/electrical industry and indoor source emissions. Factor 2
contributed the most to TBEP (78.0%), followed by TDCPP (44.7%), while it did not contribute to





TnBP. Therefore, factor 2 was deduced as the food/cosmetics industry and traffic emissions. Factor 3
contributes 71.7% of the total TnBP, which can be deduced as chemical industrial source.

## 4. Conclusions and Implications

Compared to levels of OPEs in other cities, the levels of OPEs measured in this study were comparable
or even higher than most of other studies. This suggests that during the shift of labour-intensive
manufacturing from the coastal developed areas to inland regions, OPEs were widely used in industrial
and manufacturing processes in southwest China which needs concern.
This intensive sampling campaign of urban and suburban area found no significant spatial variability of
$\Sigma_7$OPEs across Chengdu, China, but the most notable seasonal variation was observed for alkyl
phosphate, followed by halogenated OPEs and aryl OPEs. Higher concentrations and more dispersed
pattern of OPEs in autumn/winter than that in summer, with TBEP, TCEP, TCPP and TnBP being the
dominant compounds. The occurance of unexpected high level of individual OPEs at different sites in
autumn might indicate that there was a noteworthy emission. PCA analysis showed the main sources of
OPEs in $PM_{2.5}$ include plastic industry/interior decoration /traffic emission (34.5%) and chemical,
mechanical and electrical industry (27.8%). PMF showed the main sources were the plastics/electrical
industry and indoor source emissions. OPEs have a wide range of physical and chemical properties,
combined with differences in its behavior identified in this study, the management of OPEs as
individual compounds instead of a single chemical class should be considered. In addition, due to the
special topography and meteorological conditions of the inland city, the distribution and seasonal
variation of OPEs in the air in this study were significantly different from that of most coastal cities
and over the sea. The sustained and stable high local emissions are particularly noteworthy. The
chlorinated phosphate, especially TCPP and TCEP, which are highly toxic and not easy to degrade in
the environment, have a high content. Their usage and source emissions should be controlled.

## Acknowledgments

We acknowledge financial support from National Natural Science Fund (41773072, 21407014,

394     41831285).



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



Figure Captions:
Fig.1. Levels and seasonal variation of $\Sigma_7$OPEs at each sampling site. A:autumn, W:winter, Sp:spring,
Su:summer, Sub:suburbs, Dow:dowtown, S:south, E:east, N:north, W:west.
Fig. 2 Seasonal variation of $\Sigma_7$OPEs at each sampling site.
Fig.3 The seasonal variation of monomer OPEs in Chengdu city. A:Autumn, W:Winter, Sp:Spring,
Su:Summer, Sub:Suburbs, Dow:Dowtown, S:South, E:East,N:North, W:West
Fig.4 Relationship of OPE monomer concentration in PM$_{2.5}$ and its vapor pressure
Fig. 5 Spearman's ranks correlation coefficients between the concentrations of individual OPEs in
PM$_{2.5}$ samples
Fig. 6 Relationship between OPEs in atmospheric PM$_{2.5}$ and in soil.
Table Captions:
Table 1 Table 1 The annual median concentrations of OPEs in PM$_{2.5}$ form Chengdu (ng m$^{-3}$).
Table 2 The correlation analysis of monomer OPEs in downtown and suburb sampling sites.
*. Correlation is significant at the 0.05 level (2-tailed).
**. Correlation is significant at the 0.01 level (2-tailed).


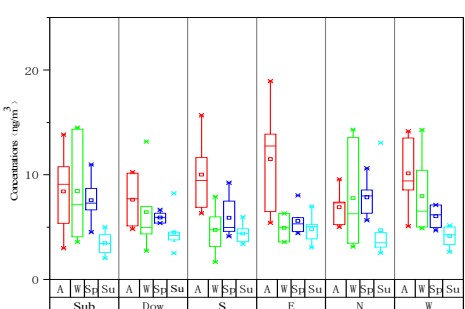


**Fig.1 Levels and seasonal variation of Σ7OPEs at each sampling site. A:autumn, W:winter, Sp:spring,**


**Su:summer, Sub:suburbs, Dow:dowtown, S:south, E:east, N:north, W:west.**


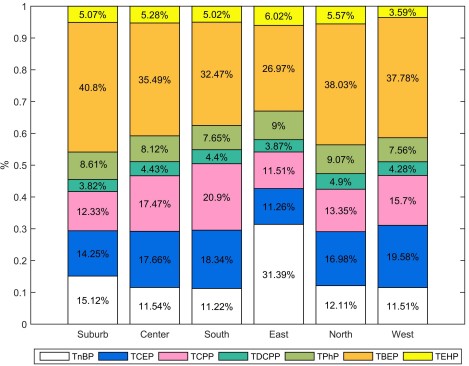


**Fig.2 Seasonal variation of Σ7OPEs at each sampling site.**



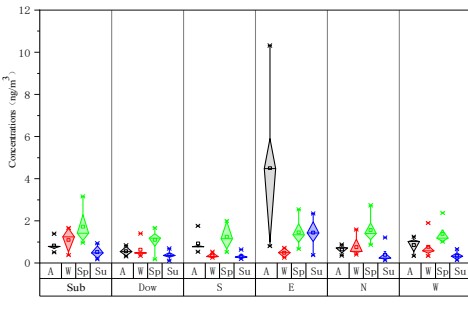

TnBP

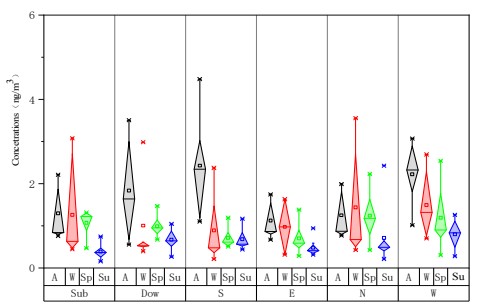

TCEP



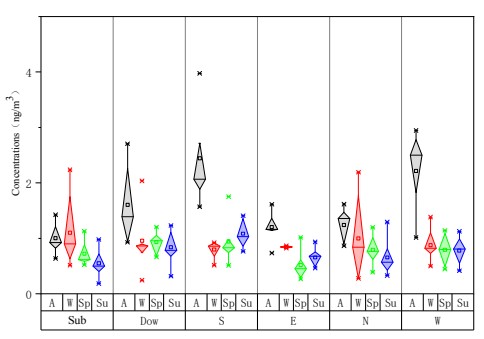

TCPP

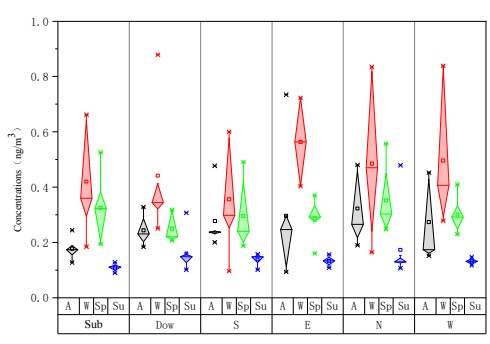

TDCPP

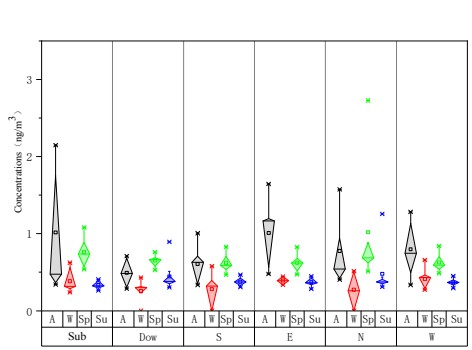

TPhP

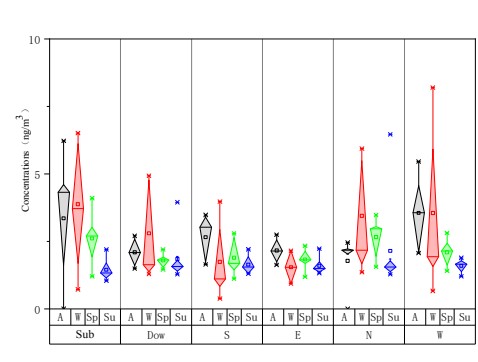

TBEP

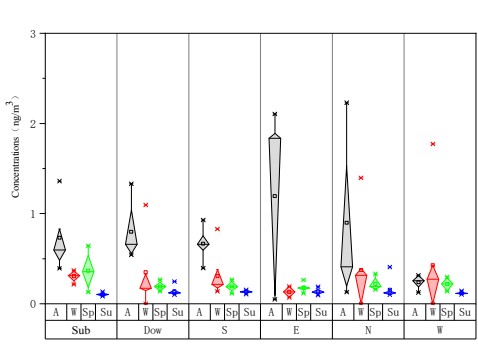

TEHP



**Fig.3 The seasonal variation of monomer OPEs in Chengdu city. A:Autumn, W:Winter, Sp:Spring,**
**Su:Summer, Sub:Suburbs, Dow:Dowtown, S:South, E:East,N:North, W:West.**

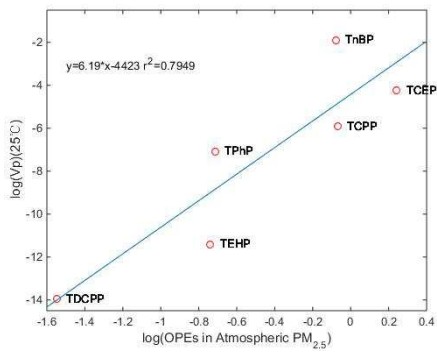


**Fig.4 Relationship of OPE monomer concentration in PM2.5 and its vapor pressure**

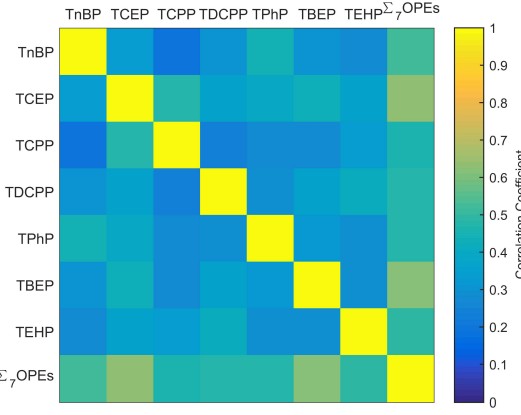





**Fig.5 Spearman's ranks correlation coefficients between the concentrations of individual OPEs in PM2.5**
**samples**

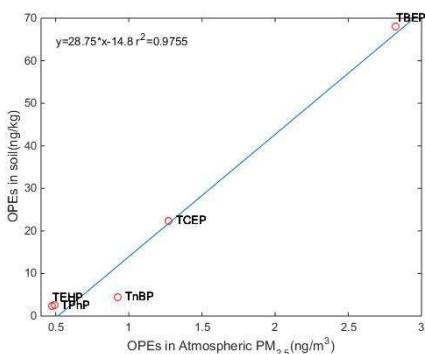


**Fig.6 Relationship between OPEs in atmospheric PM2.5 and in soil.**

**Table 1 The annual median concentrations of OPEs in PM2.5 form Chengdu (ng m$^{-3}$).**

| Orientation | TnBP | TCEP | TCPP | TDCPP | TPhP | TBEP | TEHP | $\Sigma_7$OPEs |
|---|---|---|---|---|---|---|---|---|
| suburb | 1.0 | 1.0 | 0.8 | 0.3 | 0.6 | 2.7 | 0.3 | 6.7 |
| downtown | 0.7 | 1.0 | 1.0 | 0.3 | 0.5 | 2.1 | 0.3 | 5.8 |
| south | 0.7 | 1.1 | 1.2 | 0.3 | 0.5 | 1.9 | 0.3 | 5.9 |
| east | 2.1 | 0.8 | 0.8 | 0.3 | 0.6 | 1.8 | 0.4 | 6.6 |
| north | 0.8 | 1.1 | 0.9 | 0.3 | 0.6 | 2.5 | 0.4 | 6.7 |
| west | 0.8 | 1.4 | 1.1 | 0.3 | 0.5 | 2.6 | 0.3 | 6.9 |
| median | 1.0 | 1.1 | 1.0 | 0.3 | 0.5 | 2.3 | 0.3 | 6.4 |


**Table 2 The correlation analysis of monomer OPEs in downtown and suburb sampling sites.**

| | | TnBP | TCEP | TCPP | TDCPP | TPhP | TBEP | TEHP |
|---|---|---|---|---|---|---|---|---|
| Downtown | TnBP | 1 | .408[*] | 0.319 | 0.15 | .455[*] | 0.187 | 0.105 |
| | TCEP | .408[*] | 1 | .818[**] | 0.165 | 0.342 | .447[*] | .449[*] |
| | TCPP | 0.319 | .818[**] | 1 | 0.184 | 0.392 | .447[*] | .500[*] |
| | TDCPP | 0.15 | 0.165 | 0.184 | 1 | 0.053 | 0.216 | 0.175 |
| | TPhP | .455* | 0.342 | 0.392 | 0.053 | 1 | 0.104 | -0.081 |





| | | | | | | | |
|---|---|---|---|---|---|---|---|
| | TBEP | 0.187 | .447* | .447* | 0.216 | 0.104 | 1 | 0.338 |
| | TEHP | 0.105 | .449* | .500* | 0.175 | -0.081 | 0.338 | 1 |
| Suburb | TnBP | 1 | .566** | .476* | .650** | 0.269 | .417* | 0.141 |
| | TCEP | .566** | 1 | .852** | .683** | 0.368 | .784** | .423* |
| | TCPP | .476* | .852** | 1 | .686** | 0.304 | .701** | 0.297 |
| | TDCPP | .650** | .683** | .686** | 1 | 0.175 | .708** | 0.158 |
| | TPhP | 0.269 | 0.368 | 0.304 | 0.175 | 1 | .512** | .629** |
| | TBEP | .417* | .784** | .701** | .708** | .512** | 1 | .434* |
| | TEHP | 0.141 | .423* | 0.297 | 0.158 | .629** | .434* | 1 |

*. Correlation is significant at the 0.05 level (2-tailed).
**. Correlation is significant at the 0.01 level (2-tailed).