# Peer review of "Measurement report: Seasonality, distribution and sources of organophosphate esters in PM2.5 from an inland urban city in Southwest China"

_Atmospheric Chemistry and Physics, 2020_

## Referee Comment (RC1) · Anonymous Referee #1 · 3 Jun 2020

Reviews: The manuscript reported the measurement of OPEs in PM2.5 in Chengdu, China and presented the seasonal and spatial distributions, and the potential sources of the OPEs by using multiple correlation tests. The analysis and reported data were consistent with the conclusions. The measurements and findings are critical to fill in the knowledge gap of OPEs levels in inland cities. However, several issues need to be addressed before acceptance for publication. Besides some typos and wording changes, Figure 2 seems not matching the context since no seasonal variations can be seen. Since different statistical tests were used, e.g. Pearson correlation test, spearman's rank correlation test, and nonparametric test, a clearer statement of conditions (e.g. normality check) to use those tests is needed. Lastly, the references need to be checked carefully since some of them are either not matched or not cited appropriately.

Specific comments on the manuscript

1. Introduction: line 30, the reference "Bacoloni, A. et al. 2008" was wrongly matched, since the referenced study measured water samples instead of air.

2. Introduction: line 32, the reference "Araki et al. 2014" didn't measured organisms, instead, they measured dust.

3. Introduction: line 34, the reference "Matthews, et al., 1990, 1993". Both references are animal studies. Thus, stating "many scholars found that OPEs have negative effects on the human body..." is not appropriate.

4. Introduction: line 41, the reference " Covaci et al. 2007" focused on analytical method development instead of measurement reports, not sure if it is a good reference here.

5. Introduction: line 53, change "14335" to "14,335".

6. Materials and Methods: line 72, (Sigma Aldrich, ? location? country?); Be consistent in the text in terms of listing instrument/chemical manufacturing info.

7. Results: line 124, "heavy or light polluted area" may be better.

8. Results: line 126-128, rephrase the sentence to make it more precise.

9. Results: line 136, "And they were lower than".

10. Results: line 138, add a space before (Wang, T. et al.), Double check other places in the text to make the format consistent.

11. Results: section 3.3. "Seasonal and spatial variations of OPEs in PM2.5", starting line 186, there is a mis-match in Fig.2 with the context. Where are the seasonal

variations presented in Fig.2? Only site variations were presented here.

12. Results: line 227, delete first "the". "Considering" instead of "Considered".

13. Results: line 228, 229, lowercase "the third ring road".

14. Results: line 229, maybe " the uniform patterns of OPEs levels and distribution across the city is understandable"?

15. Results: line 229, delete "But".

16. Results: line 232, "shoemaking industrial parks are located in the suburbs".

17. Results: line 233, "high levels".

18. Results: line 235, delete "to the individual OPE".

19. Results: line 257, 258, " their gas-particle distributions determine their concentrations in PM2.5".

20. Results: line 266, is it "Fig.4 showed" or "Fig.5 showed"?

21. Results: line 275, delete "so".

22. Results: line 282, add "The correlations between" before actually listing pairs of OPE monomers.

23. Results: line 284, delete second "was".

24. Results: section 3.4.3 "Correlation analysis of OPEs and PM2.5 concentrations", you mentioned Fig. S2, in which you used Pearson correlation tests. Why not spearman's rank correlation tests as you used in Figure 5?

25. Results: line 291, add "found" after "was".

26. Results: line 315, "different uses".

27. Results: line 338,339, add a reference to the statement "Chengdu's wind has

always been...".

28. Conclusions and Implications: line 372, "compared to the levels of OPEs in other cities".

29. Conclusions and Implications, line 390, maybe change "not easy to degrade" to "persistent"? What do you mean by "have a high content"?, change the wording to clarify.

30. Reference: line 486-488, where the reference was cited? Cannot locate it in the text "Tang, R., Keming, M.A., Zhang, Y., Mao, Q.: Health risk assessment of heavy metals of street dust in Beijing, Acta. Scientiae. Circumstantiae., 32, 2006-2015, https://doi.org/10.13671/j.hjkxxb.2012.08.029, 2012."

31. Reference: what is the novelty in this paper compared with your reference paper in Chinese (Line 512-514) "Yin, H.L., Li, S.P., Ye, Z.X., Yang, Y.C., Liang, J.F., You, J.J.: Pollution Level and Sources of 513 Organic Phosphorus Esters in Airborne PM2.5 in Chengdu City, Environ. Sci. (in chinese), 36, 3566-3572, https://doi.org/10.13227/j.hjkx.2015.10.003, 2015."

32. Reference: line 515-517, reference "Zhang, Q. H., Yang, W. N., Ngo, H. H., Guo, W. S., Jin, P. K., Dzakpasu, M.: Current status of urban wastewater treatment plants in China, Environ. Int., 92-93, 11-22, https://doi.org/10.1016/j.envint.2016.03.024, 2016" might not be a good reference to be used here.

33. Figure 2: where is the seasonal variations? As only site variation is presented here.

34. Figure 4: line 542, be consistent with your notations/subscripts in the manuscript, PM2.5 or PM2.5. Same issue in line 544 etc.

35. Figure 5: Line 544, Should be "Spearman's rank correlation coefficients". Double check other places to be consistent.

36. Table 1: line 549, "orientation" of what? wind direction? If so, may want to use a different term since suburb and downtown probably do not quite fit.

37. In Figure 5 "Spearman's ranks correlation coefficients between the concentrations of individual OPEs in PM2.5 samples" and Figure S2 "Scatter plot of OPEs and PM2.5", spearman's rank tests and Pearson's correlation coefficients were used. Could you explain more about the selection of two different correlation tests?

---

## Referee Comment (RC2) · Anonymous Referee #2 · 5 Jun 2020

Thanks for the invitation to review. I read the manuscript by Yin et al. with interest. The authors reported concentrations of seven OPEs in PM2.5 from Chengdu, China, tracked their possible sources, and conducted source apportionment using PCA and the PMF receptor model. My utmost concern is the data accuracy as some required QA/QC procedures were missing. Additionally, the manuscript is a little hard to read as it has a number of grammatical issues, and several statements lacked reference supports. Though this study provided a few useful information (e.g., difference in OPE profiles between inland and costal cities), its novelty and quality at current version

may not be sufficient enough for the Atmospheric Chemistry and Physics. My specific comments are as follows:

Major concern: Novelty: There is a similar study previously conducted by the leading author here. What makes this manuscript distinct from that previous one? Authors should elaborate more the novelty of this study. QA/QC: 1) As no surrogate standards were spiked prior to sample treatment, how did authors evaluate OPE recoveries from the analytical procedures?; 2) How was the matrix effect assessed and compensated?; and 3) The data from field blanks were missing. PMF model: How were the uncertainties determined? Which references were referred to for identification of sources associated with each factor? I also want to see the source profile of each factor.

Minor concern: Line 8: "emerging contaminants" → "contaminant of emerging concern". OPEs have been produced for decades. Line 9: "centers" → "areas" Line 13: "...which TOGETHER made up..." Line 18: OPEs can transfer from soil to air particles via suspension and volatilization as well. Actually, authors mentioned this at Lines 303-304. Lines 32-35: A weird sentence, please rephrase it. Line 35: Reference is needed for the "OPE restrictions". Lines 38-39: Reference is needed. Line 45: Which type of matrix is referred to for "Concentrations of OPEs in most cities..." I looked at the references cited, but not all of them talked about PM2.5. Lines 54-56: How about "Chengdu is an important city in Southwest China due to its role as a national high-tech industrial base, a commercial logistics center, and a comprehensive transportation hub"? Line 82: Sampling intervals? Line 86: Was the analytical method used here applied in any previous studies? Lines 93-94: How about "The latter eluate was collected and concentrated by vacuum-condensing..."? Lines 114-118: Could concisely say "detected in virtually all the samples". Lines 120-121: Did "The average value...four seasons" mean "annual average level"? Line 141: Rephrase the first sentence. Lines 143-145: Explain the meaning of values in the parentheses. Lines: 165-167: References? Lines 182-184: A recent study measuring an extended list of OPEs in the Great Lakes atmosphere also found that alkyl OPEs dominated OPE compositional profiles of urban

air collected from Chicago and Cleveland (Wu et al. 2020; 10.1021/acs.est.9b07755). Line 208-210: OPE levels can be surely affected by temperature, so I suppose the authors would like to say "seasonal variations in OPE levels". Additionally, would meteorological parameters other than temperature result in the seasonal variations found in the present study? Lines 236-238: Has been mentioned before. Lines 238-248: Out of place here. Could be moved to section 3.1. Line 257: Need reference to support "they tend to be adsorbed in PM 2.5". Line 315: Other factors may lead to such difference between indoor and outdoor OPEs. For example, TBEP has the shortest atmospheric half-lives, which may explain why its dominance in indoor samples was not observed for the outdoor counterparts. Lines 350-356: References are required for identification of possible sources associated with each factor.

---

## Author Comment (AC1) · 15 Jun 2020

Thank you for your valuable comments and good advice on improving our manuscript. We are so sorry that the manuscript has some mistakes. The typos and wording of the manuscript, as well as the specific contents and references of the manuscript, have been revised as follows according to your comments. Specific comments on the manuscript 1. Introduction: line 30, the reference "Bacoloni, A. et al. 2008" was wrongly matched, since the referenced study measured water samples instead of air. Response: The reference "Bacoloni,A. et al. 2008" has been replaced by

"Guo et all., 2016" and "Li et al., 2017". Guo, Z. M., Liu, D., Shen, K.J., Li, J. Yu, Z.Q. Zhang, G.: Concentration and seasonal variation of organophosphorus flame retardants in PM2.5 of Taiyuan City, China, Earth and environment (in chinese)., 44, 600-604. https://doi.org/10.14050/j.cnki.1672-9250.2016.06.002, 2016. Li, J., Xie, Z., Mi, W., Lai, S., Tian, C., Emeis, K.C.: Organophosphate esters in air, snow and seawater in the north atlantic and the arctic, Environ. Sci. Technol., 51, 6887-6896. https://doi.org/10.1021/acs.est.7b01289, 2017. 2. Introduction: line 32, the reference "Araki et al. 2014" didn't measured organisms, instead, they measured dust. Response: "Araki et al. 2014" was deleted. 3. Introduction: line 34, the reference "Matthews, et al., 1990, 1993". Both references are animal studies. Thus, stating "many scholars found that OPEs have negative effects on the human body. . ." is not appropriate. Response: Thank you very much for your advice .The word "organisms" was replaced by "human body". 4. Introduction: line 41, the reference " Covaci et al. 2007" focused on analytical method development instead of measurement reports, not sure if it is a good reference here. Response: It's regret that the manuscript has such a mistake. This reference was deleted and other references in the manuscript have been verified. 5. Introduction: line 53, change "14335" to "14,335". Response: Thank you very much for your advice in such a detail. It has been revised to "14,335". 6. Materials and Methods: line 72, (Sigma Aldrich, ? location? country?); Be consistent in the text in terms of listing instrument/chemical manufacturing info. Response: Sigma Aldrich is the reagent production company. The manufacturing information of instruments and reagents has been indicated in the manuscript, and the full text has been checked. 7. Results: line 124, "heavy or light polluted area" may be better. Response: Thanks for your advice. "polluted" has been revised to "pollution" in the manuscript. 8. Results: line 126-128, rephrase the sentence to make it more precise. Response: The sentence has been revised to: "These data were quite consistent with our previous study which studied the annual median concentration of OPEs in PM2.5 from December 2013 to October 2014 (Yin et al., 2015). Interestingly, the concentration of $\Sigma$7OPEs at the suburban site was similar to, or even higher than some

urban sites, which indicated more local sources of these compounds in the suburban area." 9. Results: line 136, "And they were lower than". Response: The grammatical problems in this manuscript have been carefully corrected. 10. Results: line 138, add a space before (Wang, T. et al.), Double check other places in the text to make the format consistent. Response: The typos of the manuscript have been proofread. 11. Results: section 3.3. "Seasonal and spatial variations of OPEs in PM2.5", starting line 186, there is a mis-match in Fig.2 with the context. Where are the seasonal variations presented in Fig.2? Only site variations were presented here. Response: We are so sorry for this mistake. Because the version we uploaded to the website is different from the first draft, the sequence numbers of figures have been adjusted. We forgot to change it here. Figure 1 refers to "levels and seasonal variation of $\Sigma$7OPEs at each sampling site". "Figure 2" in line 186 has been changed to "Figure 1". 12. Results: line 227, delete first "the". "Considering" instead of "Considered". Response: Thank you for your valuable comments. They were revised as suggested. 13. Results: line 228, 229, lowercase "the third ring road". Response: Thanks a lot. The "third Ring Road" has been revised to "the third ring road". 14. Results: line 229, maybe "the uniform patterns of OPEs levels and distribution across the city is understandable"? Response: This sentence has been revised to "the similar level and profile of OPEs across the city was understandable". 15. Results: line 229, delete "But". Response: "But" has been deleted. 16. Results: line 232, "shoemaking industrial parks are located in the suburbs". Response: This sentence has been revised to "...shoemaking industrial parks are located in the suburbs". Thank you for your advice. 17. Results: line 233, "high levels". Response: It was revised as suggested. 18. Results: line 235, delete "to the individual OPE". Response: "to the individual OPE" has been deleted. Thank you for pointing out this problem. 19. Results: line 257, 258, " their gas-particle distributions determine their concentrations in PM2.5". Response: Thank you for your correction. "...distributions determines" has been revised to "distributions determine". 20. Results: line 266, is it "Fig.4 showed" or "Fig.5 showed"? Response: We are so sorry for this mistake. Line 266 should be "Figure 5 showed" and it was revised in the

manuscript. 21. Results: line 275, delete "so". Response: "So" has been deleted. Thanks for your advice. 22. Results: line 282, add "The correlations between" before actually listing pairs of OPE monomers. Response: "The relationships between" has been added. 23. Results: line 284, delete second "was". Response: The second "was" has been deleted. Thank you for reminding us of this mistake 24. Results: section 3.4.3 "Correlation analysis of OPEs and PM2.5 concentrations", you mentioned Fig. S2, in which you used Pearson correlation tests. Why not spearman's rank correlation tests as you used in Figure 5? Response: As we know, Pearson evaluates the linear relationship between the two variables, while Spearman evaluates the monotonic relationship between the two variables. According to the results of other literature (Wong et al., 2018) and our hypothesis, we think that PM2.5 concentration is linearly related to the content of OPEs. So we carried out Pearson correlation tests in Fig. S2 according to the hypothesis. The results showed that the correlation was very poor, which was totally different from what we expected. In order to emphasize the difference of correlation between OPEs/other pollutants and PM2.5 concentration, Pearson correlation test result was used. 25. Results: line 291, add "found" after "was". Response: "found" has been added. 26. Results: line 315, "different uses". Response: "us" has been revised to "uses" 27. Results: line 338,339, add a reference to the statement "Chengdu's wind has always been. . .". Response: Thanks for your advice. A website was added. "Chengdu is a city located in the interior of China" has been added to illustrate that its wind intensity is smaller than coastal cities. ïijĹhttps://baike.baidu.com/item/%E6%88%90%E9%83%BD/128473?fr=aladdinïijĽ
28. Conclusions and Implications: line 372, "compared to the levels of OPEs in other cities". Response: Line 372 has been changed to "compared to the levels of OPEs in other cities" 29. Conclusions and Implications, line 390, maybe change "not easy to degrade" to "persistent"? What do you mean by "have a high content"?, change the wording to clarify. Response: The sentence in line 390 has been changed to "the chlorinated phosphate, especially TCPP and TCEP, which are highly toxic and persistent in the environment, have high concentrations in this

study." 30. Reference: line 486-488, where the reference was cited? Cannot locate it in the text "Tang, R., Keming, M.A., Zhang, Y., Mao, Q.: Health risk assessment of heavy metals of street dust in Beijing, Acta. Scientiae. Circumstantiae., 32, 2006-2015, https://doi.org/10.13671/j.hjkxxb.2012.08.029, 2012." Response: It has been deleted. 31. Reference: what is the novelty in this paper compared with your reference paper in Chinese (Line 512-514) "Yin, H.L., Li, S.P., Ye, Z.X., Yang, Y.C., Liang, J.F., You, J.J.: Pollution Level and Sources of 513 Organic Phosphorus Esters in Airborne PM2.5 in Chengdu City, Environ. Sci. (in chinese), 36, 3566-3572, https://doi.org/10.13227/j.hjkx.2015.10.003, 2015." Response: The article we published earlier is a report of our experiment results from only two sampling sites. The purpose of that paper was to report the pollution level and distribution of the atmospheric OPEs at urban and suburban sites. Interestingly, we found the seasonal variations of OPEs were significantly different from PM2.5 concentrations and PM2.5-bound PAHs, etc.. So we carried out a more detailed experiment with six sampling sites in the second year. In this paper, except for reporting the level and seasonal variations of OPEs at six sites, we paid more attention to investigate the relationships and correlations among the target compounds or with influence factors and illustrate the potential sources of OPEs in PM2.5. For example, whether different functional areas affect the distributions of atmospheric OPEs, correlations of OPEs with environmental factors (vapor pressure, boiling points, etc.), correlations of OPEs with PM2.5 concentrations, correlations of OPEs in PM2.5 and soil, correlations of OPEs in indoor and outdoor air were all discussed. These differences are the innovation of this paper. 32. Reference: line 515-517, reference "Zhang, Q. H., Yang, W. N., Ngo, H. H., Guo, W. S., Jin, P. K., Dzakpasu, M.: Current status of urban wastewater treatment plants in China, Environ. Int., 92-93, 11-22, https://doi.org/10.1016/j.envint.2016.03.024, 2016" might not be a good reference to be used here. Response: Thanks for your advice. It has been deleted. 33. Figure 2: where is the seasonal variations? As only site variation is presented here. Response: There are some errors in the arrangement of the sequence number of the figure. Figure 1 refers to "levels and seasonal variation

of $\Sigma$7OPEs at each sampling site", and Figure 2 refers to "the proportion of individual OPE in the $\Sigma$7OPEs at each point". Figure 2 in the text should actually be Figure 1. 34. Figure 4: line 542, be consistent with your notations/subscripts in the manuscript, PM2.5 or PM2.5. Same issue in line 544 etc. Response: Thanks for your advice. All "PM2.5" appearing in the manuscript has been replaced by "PM2.5". 35. Figure 5: Line 544, Should be "Spearman's rank correlation coefficients". Double check other places to be consistent. Response: It has been revised to "Spearman's rank correlation coefficients". We have checked other places throughout the manuscript. 36. Table 1: line 549, "orientation" of what? wind direction? If so, may want to use a different term since suburb and downtown probably do not quite fit. Response: "Orientation" refers to the direction of the city, not the wind direction. It has been replaced with "sampling sites". 37. In Figure 5 "Spearman's ranks correlation coefficients between the concentrations of individual OPEs in PM2.5 samples" and Figure S2 "Scatter plot of OPEs and PM2.5", spearman's rank tests and Pearson's correlation coefficients were used. Could you explain more about the selection of two different correlation tests? Response: Pearson evaluates the linear relationship between the two variables, while Spearman evaluates the monotonic relationship between the two variables. According to the results of other literatures and our hypothesis, we selected the different test method. In addition, when choosing which of the two test methods to use, firstly we would use the data distribution map to determine whether the data was normal distribution or non normal distribution. If it was normal distribution, Pearson's correlation coefficients were used. If not, Spearman's ranks correlation coefficients were used.

---

## Author Comment (AC2) · 23 Jun 2020

Response to Anonymous Referee #2 Dear sir, we gratefully thanks for the precious time the reviewer spent making constructive remarks and totally understand the reviewer's concern. The pre-experiment was carried out before the experiment. We conducted the thorough experiment for the quality control and quality assurance including the blank experiment, recoveries of internal standard (TDCPP-d15 and TPhP-d15) in samples for evaluating the accuracy, blank experiment (field blanks, solvent blanks, matrix blanks), precision experiment, etc.. Due to the limited space of the paper, and the focus

of this paper is not on the establishment of analytical methods, it is simplified a lot in the QA/QC part. But we have done all the related experiments for QA/QC, and the results were good. In the revised manuscript, we have added them in QA/QC part. Therefore, there is no need to worry about the accuracy of the data. But it's a pity that there were many grammatical problems and reference problems in the manuscript. We all corrected them and sincerely hope that the manuscript can meet the requirements of Atmospheric Chemistry and Physics after modification. According to your constructive comments, the revisions of the manuscript are as follows:

Major concern: Novelty: There is a similar study previously conducted by the leading author here. What makes this manuscript distinct from that previous one? Authors should elaborate more the novelty of this study. Response: The article we published earlier is a report of our experiment results from only two sampling sites. The purpose of that paper was to report the pollution level and distribution of the atmospheric OPEs at urban and suburban sites. Interestingly, we found the seasonal variations of OPEs were significantly different from PM2.5 concentrations and PM2.5-bound PAHs, etc.. So we carried out a more detailed experiment with six sampling sites in the second year. In this paper, except for reporting the level and seasonal variations of OPEs at six sites, we paid more attention to investigate the relationships and correlations among the target compounds or with influence factors and illustrate the potential sources of OPEs in PM2.5. For example, whether different functional areas affect the distributions of atmospheric OPEs, correlations of OPEs with environmental factors (temperature, wind, vapor pressure, boiling points, etc.), correlations of OPEs with PM2.5 concentrations, correlations of OPEs in PM2.5 and soil, correlations of OPEs in indoor and outdoor air were all discussed. These differences are the innovation of this paper.

QA/QC: 1) As no surrogate standards were spiked prior to sample treatment, how did authors evaluate OPE recoveries from the analytical procedures? Response: "Thorough QA/QC procedures for OPE analysis were conducted to ensure data quality. To evaluate the recovery efficiencies of analytical procedures, all samples were added

with internal standard (TDCPP-d15 and TPhP-d15), and the accuracy was evaluated by their recoveries. The concentrations of the 7 OPEs were determined by an external standard method. The correlation coefficients of the standard curves of the seven OPE monomers were all greater than 0.990. The recoveries of 7 OPEs and the internal standard were between 78.9% and 122.5%." was added in the manuscript. There are only two internal standards, so we use them to ensure the recovery, but use external standard method to quantify the target compounds. In addition, a matrix blank was run in parallel with every batch of samples for the analysis of OPEs. Only TnBP was detected in the blanks, and the level of TnBP found in the blanks was <5% of the concentrations measured in all samples, which means it was negligible. Field blanks were done at each site to evaluate the background contamination in the field. TBEP, TnBP and TEHP were detected in it. The level of them found in the blank were <15% of the concentrations measured in all samples. The correlation coefficients of the standard curves of the seven OPE profiles were all greater than 0.990.These all could ensure the accuracy of the data.

2) How was the matrix effect assessed and compensated? Response: The matrix effect was assessed by the matrix blank experiment. The blank quartz membrane was added with the internal standard (TDCPP-d15 and TPhP-d15) and OPEs standard. After the whole pretreatment process, the recoveries of 7 OPEs and the internal standard were all between 70% and 120%. So the data was not corrected and the matrix effect was not compensated.

3) The data from field blanks were missing. PMF model: How were the uncertainties determined? Which references were referred to for identification of sources associated with each factor? I also want to see the source profile of each factor. Response: The field blanks were done which were prepared and installed in the same manner as the regular samples but without turning on the sampler motor. Due to the limited space of the paper, and the focus of this paper is not on the establishment of analytical methods, so it is simplified a lot in the QA/QC part. But we have done all the related experiments

for QA/QC, and the results were good. "Field blanks were done at each site to evaluate the background contamination in the field." TBEP, TnBP and TEHP were detected in it. The level of them found in the blank were <15% of the concentrations measured in all samples." was added in the revised manuscript. In PMF, the uncertainty is estimated by three methods: BS, disp and bs-disp. The results are shown in the table below: DISP results showed that the solution was stable because no swaps were present. BS results showed that mapping over 80% of the factors indicated that the BS uncertainties could be interpreted and the number of factors may be appropriate. All of the "Strong" species were selected for the BS-DISP error estimation. The number of DISP and BS-DISP swaps was zero. BS-DISP highlight that the solution may be reliable due to there was no swaps across two factors. Error estimation summary results BS-DISP Diagnostics: # of Cases Accepted: 100 % of Cases Accepted: 100% Largest Decrease in Q: -0.150999993 %dQ: -0.067824623 # of Decreases in Q: 0 # of Swaps in Best Fit: 0 # of Swaps in DISP: 0 Swaps by Factor: 0 0 0 DISP Diagnostics: Error Code: 0 Largest Decrease in Q: -0.005 %dQ: -0.002245848 Swaps by Factor:  0 0 0 BS Mapping: Factor 1 Factor 2 Factor 3 Unmapped Boot Factor 1 100 0 0 0 Boot Factor 2 0 100 0 0 Boot Factor 3 0 0 100 0

The source profile of each factor: Factor Profiles (% of species sum) from Base Run (Convergent Run) TnBP TCEP TCPP TDCPP TPhP TBEP TEHP Factor 1 28.69 70.95 70.72 31.01 58.34 0.00 70.93 Factor 2 0.00 20.31 25.47 44.72 13.97 77.95 26.41 Factor 3 71.31 8.73 3.81 24.27 27.69 22.05 2.66

The references were referred to for identification of sources associated with each factor. "Factor 1 was deduced to be the plastics/electrical industry and indoor source emissions (Esch, 2000; Leisewitz et al., 2000; Stevens et al., 2006). Factor 2 contributed the most to TBEP (78.0%), followed by TDCPP (44.7%), while it did not contribute to TnBP. Therefore, factor 2 was deduced as the food/cosmetics industry and traffic emissions (Marklund et al., 2005). Factor 3 contributes 71.7% of the total TnBP, which can be deduced as chemical industrial source (Regnery et al., 2011). "

Minor concern: Line 8: "emerging contaminants"→"contaminant of emerging concern". OPEs have been produced for decades. Response: We're sorry for the improper expression. This expression has been revised to " OPEs are a kind of contaminants of emerging concern in recent years"

Line 9: "centers"→ "areas" Response: Thanks for your advice. "Centers" has been replaced by "areas".

Line 13... which TOGETHER made up..." Response: Thanks for your advice. It has been revised to "which together made up".

Line 18: OPEs can transfer from soil to air particles via suspension and volatilization as well. Actually, authors mentioned this at Lines 303-304. Response: Thanks for your advice. "suggested the atmospheric PM2.5 settlement is an important source of OPEs in soil" has been deleted in line 18.

Lines 32-35: A weird sentence, please rephrase it. Response: Thanks for your advice. After rephrasing, the sentence becomes "However, many scholars found that the residues of OPEs in the environment could cause toxic effects on organisms. (WHO, 1991, 1998, 2000; Kanazawa et al., 2010; Van der Veen and de Boer, 2012; Du et al., 2015)".

Line 35: Reference is needed for the "OPE restrictions". Response: Three references were added for reference: Blum, A.; Behl, M.; Birnbaum, L. S.; Diamond, M. L.; Phillips, A.; Singla, V.; Sipes, N. S.; Stapleton, H. M.; Venier, M. Organophosphate ester flame retardants: Are they a regrettable substitution for polybrominated diphenyl ethers? Environ. Sci. Technol. Lett. 2019, 6, 638-649. Exponent. California bans flame retardants in certain consumer products. 2018, Available at: https://www.exponent.com/knowledge/alerts/2018/09/california-bans-flame-retardants/?pageSize=NaN&pageNum=0&loadAllByPageSize=true (accessed February 15, 2020) State of California. Safer consumer products (SCP) information management system. 2020. Available at:

https://calsafer.dtsc.ca.gov/cms/search/?type=Chemical (accessed February 21, 2020).

Lines 38-39: Reference is needed. Response: Thanks for your advice. References have been added according to your suggestionïijŽ Möller, A.; Xie, Z.; Caba, A.; Sturm, R.; Ebinghaus, R. Organophosphorus flame retardants and plasticizers in the atmosphere of the North Sea. Environ. Pollut. 2011, 159, 3660-3665. Möller, A.; Sturm, R.; Xie, Z.; Cai, M.; He, J.; Ebinghaus, R. Organophosphorus flame retardants and plasticizers in airborne particles over the Northern Pacific and Indian Ocean toward the polar regions: Evidence for global occurrence. Environ. Sci. Technol. 2012, 46, 3127-3134. McDonough, C. A.; De Silva, A. O.; Sun, C.; Cabrerizo, A.; Adelman, D.; Soltwedel, T.;Bauerfeind, E.; Muir, D. C. G.; Lohmann, R. Dissolved organophosphate esters and polybrominated diphenyl ethers in remote marine environments: Arctic surface water distributions and net transport through Fram Strait. Environ. Sci. Technol. 2018, 52, 6208-6216.

Line 45: Which type of matrix is referred to for "Concentrations of OPEs in most cities..."I looked at the references cited, but not all of them talked about PM2.5. Response: This matrix is only for outdoor atmospheric environment. "Concentrations of OPEs in most cities..." has been revised to "Concentrations of atmospheric OPEs in most cities". Not all of the references we cited talked about PM2.5, but they were all about OPEs in atmospheric particles.

Lines 54-56: How about "Chengdu is an important city in Southwest China due to its role as a national high-tech industrial base, a commercial logistics center, and a comprehensive transportation hub"? Response: Thanks for your advice. This sentence has been added a website: https://en.wikipedia.org/wiki/Chengdu.

Line82: Sampling intervals? Response: The sampling campaign was carried out between October 2014 and September 2015. "In each season, continuous sampling was carried out for about one week, except for rainy day. In autumn, the sampling duration

was from October 23 to October 29, 2014 (no sampling due to rain on October 26 and 27); in winter, the sampling time was from December 22 to December 30, 2014 (no sampling due to rain on October 25 and 26); in spring, 2015, the sampling time was from March 25 to March 30, 2015; in summer, the sampling time was 2015 From July 16 to July 24 (no sampling due to rain on July 21). "has been added in the revised manuscript. Each collection campaign lasted 23 h. The interval of each sample was 1h.

Line 86: Was the analytical method used here applied in any previous studies? Response: Based on the references of Möller et al (2012), we established the quantitative analysis method in the laboratory. This analytical research method has been applied in our previous studies. Li, S. P.; Yin, H. L.; YE, Z. X.; Liang, J. F.; Hao, Y. F. GC-MS determination of 7 organic phosphate ester flame retardants in atmospheric particulates with chromatography purification: PTCA (Part B: Chem Anal). 2015, 051(005):581-585. Yin, H.L., Li, S.P., Ye, Z.X., Yang, Y.C., Liang, J.F., You, J.J. Pollution level and sources of organic phosphorus esters in airborne PM2.5 in Chengdu City, Environ. Sci. (in chinese), 36, 3566-3572, https://doi.org/10.13227/j.hjkx.2015.10.003, 2015. Möller, A.; Sturm, R.; Xie, Z.; Cai, M.; He, J.; Ebinghaus, R. Organophosphorus flame retardants and plasticizers in airborne particles over the Northern Pacific and Indian Ocean toward the polar regions: Evidence for global occurrence. Environ. Sci. Technol. 2012, 46, 3127-3134.

Lines 93-94: How about "The latter eluate was collected and con-centrated by vacuum-condensing..."? Response: The solvent extracts were concentrated to nearly dry by vacuum condensing equipment and then fixed volume to 200 $\mu$L with hexane. Then it was placed in a sample bottle to wait for the injection of gas chromatography-mass spectrometry (GC-MS).

Lines 114-118: Could concisely say "detected in virtually all the samples". Response: Thanks for your advice. "Four OPEs (TCPP, TDCPP, TCEP and TnBP) were detected in all samples (n=149), while TBEP was detected in all but one sample. Additionally,

TEHP was detected in 96.7% of samples overall and TPhP was detected in 98% of samples." has been revised to "Seven OPEs were found in 96.7%~100% of the samples".

Lines 120-121: Did "The average value... four seasons mean "annual average level"? Response: It means "seasonal average concentration", not "annual average level". It is so sorry that the expression is not concise enough. We have checked other places throughout the manuscript.

Line 141: Rephrase the first sentence. Response: Thanks for your advice. It has been revised to "Non-chlorinated OPEs were the predominant OPEs across Chengdu city".

Lines 143-145: Explain the meaning of values in the parentheses. Response: Thank you very much for your reminder. The expression in the parentheses has been changed to "(annual media concentration: 2.3 ng m-3, 35.3% of $\Sigma$7 OPEs)".

Lines: 165-167: References? Response: Thanks for your advice. Two company websites for producing and selling OPEs have been added: https://show.guidechem.com/hainuowei, http://www.sinostandards.net/index.php

Lines182-184: A recent study measuring an extended list of OPEs in the Great Lakes atmosphere also found that alkyl OPEs dominated OPE compositional profiles of urban air collected from Chicago and Cleveland (Wu et al. 2020; 10.1021/acs. est.9b07755). Response: Thank you for your reminder. We have referred to the results of this study. For example, (1)"Wu et al. (2020) reported that median concentrations of $\Sigma$OPEs for summer samples were up to 5 times greater than those for winter samples. The similar seasonal patterns were reported by Salamova et al. (2014) for the atmospheric particle-phase OPE concentrations in samples collected from the Great Lakes in 2012. A reasonable explanation is that OPEs are not chemically bound to the materials in which they are used and higher temperatures may facilitate their emission from buildings and vehicles." has been added in the revised manuscript. (2)"Wu et al. (2020) also reported that alkyl OPEs dominated OPE compositional profiles of urban air collected

from Chicago and Cleveland."(3) "Interestingly, in this study, alkyl OPEs dominated both urban and suburban sites. This was extremely different from the results reported by Wu et al. (2020) that alkyl OPEs dominated at urban sites, chlorinated OPEs were prevalent at rural sites, and aryl OPEs were most abundant at remote locations. " has been added in the revised manuscript.

Line 208-210: OPE levels can be surely affected by temperature, so I suppose the authors would like to say "seasonal variations in OPE levels". Additionally, would meteorological parameters other than temperature result in the seasonal variations found in the present study? Response: Thanks for your advice. Based on our experience, we also strongly agree that temperature and other meteorological factors will affect the level of pollutants in PM2.5. However, the concentration of OPEs found in this study did not varied much in the four seasons, which was significantly different from other pollutants. Some literatures showed that the seasonal variations of OPEs in some coastal cities were significantly affected by temperature (Liu et al., 2016; Wang et al., 2019). For example, Wang et al. (2019) reported seasonality was discovered for OPEs in both gas phase and PM2.5 with their concentrations higher in hot seasons in Dalian, which may due to the temperature-driven emission or gas-particle partitioning. However, "In our study, the correlation analysis between the temperature, wind speed, wind direction and $\Sigma$7OPEs concentrations has been done. The results showed statistically significant negative correlations between temperature and $\Sigma$7OPEs (R= -0.355, p<0.01). The lowest concentrations of $\Sigma$7OPEs and individual compound were observed in summer suggesting the OPEs level was not driven by the temperature-driven emission. Gas-particle partitioning and local emission sources may contribute to the variation." These have been added in the revised manuscript. In addition, other meteorological parameters with high contributions to the seasonal variations were not found in the present study.

Lines 236-238: Has been mentioned before. Lines 238-248: Out of place here. Could be moved to section 3.1. Response: Thanks for your advice. Lines 236-238 have been

deleted.

Line 257: Need reference to support "they tend to be adsorbed in PM 2.5". Response: Thanks for your advice. References have been added: (Wang et al., 2019) Wang, Y., Bao. M. J., Tan. F., Qu. Z. P., Zhang. Y. W., Chen. J. W.: Distribution of organophosphate esters between the gas phase and PM2.5 in urban Dalian, China, Environ. Pollut., https://doi.org/10.1016/j.envpol.2019.113882, 2019.

Line 315: Other factors may lead to such difference between indoor and outdoor OPEs. For example, TBEP has the shortest atmospheric half-lives, which may explain why its dominance in indoor samples was not observed for the outdoor counterparts. Response: Of course, other factors may also cause differences in the content of indoor and outdoor OPEs. The reasons for the difference in indoor and outdoor OPEs content have been supplemented and improved as follows: "Except for the different usage of OPEs, many factors may also lead to differences between indoor and outdoor OPEs. For example, TBEP has the shortest atmospheric half-lives, which may explain why its dominance in indoor samples was not observed for the outdoor counterparts. Studies in Swedish (Wong et al., 2018) reported the concentrations of OPEs in indoor air were TCPP > TCEP > TBEP > TnBP> TPhP, and in outdoor urban air were TBEP > TCPP > TCEP > TnBP > TPhP (Wong, 2018) which also indicated the differences of OPEs profile in indoor and outdoor air. They found that activities in the building, e.g. floor cleaning, polishing, construction, introduction of new electronics and changes in ventilation rate could be key factors in controlling the concentration of indoor air pollutants, while the observed seasonality for OPEs in outdoor air was due to changes in primary emission."

Lines 350-356: References are required for identification of possible sources associated with each factor. Response: Thanks for your advice. References have been added in the manuscript. "Factor 1 can represent the sources of OPEs from the plastic industry, interior decoration and traffic emission, with the contribution ratio of 34.5% (Marklund et al., 2005; Regnery et al., 2011; CEFIC, 2002). Factor 2 has higher load

on TnBP, TEHP and TPhP. The highest load was on TnBP, which is often used as a high-carbon alcohol defoamer, mostly in industries that do not come in contact with food and cosmetics, as well as in antistatic agents and extractants of rare earth elements. TEHP can be used as an antifoaming agent, hydraulic fluid and so on. TPhP is typically used in electrical and electronic products, or plastic film and rubber (Esch, 2000; Leisewitz et al., 2000; Stevens et al., 2006; Wei et al., 2015)."

---

## Author Response (AR1)

[revised manuscript text omitted]

**Author's Response**

| Response to reviewer 1#: | |
|---|---|
| Questions: | Response: |
| The manuscript reported the measurement of OPEs in $PM_{2.5}$ in Chengdu, China and presented the seasonal and spatial distributions, and the potential sources of the OPEs by using multiple correlation tests. The analysis and reported data were consistent with the conclusions. The measurements and findings are critical to fill in the knowledge gap of OPEs levels in inland cities. However, several issues need to be addressed before acceptance for publication. Besides some typos and wording changes, Figure 2 seems not matching the context since no seasonal variations can be seen. Since different statistical tests were used, e.g. Pearson correlation test, spearC1 ACPD Interactive comment Printer-friendly version Discussion paper man's rank correlation test, and nonparametric test, a clearer statement of conditions (e.g. normality check) to use | Thank you for your valuable comments and good advice on improving our manuscript. We are so sorry that the manuscript has some mistakes. The typos and wording of the manuscript, as well as the specific contents and references of the manuscript, have been revised as follows according to your comments. |

| | |
|---|---|
| those tests is needed. Lastly, the references need to be checked carefully since some of them are either not matched or not cited appropriately. | |
| 1. Introduction: line 30, the reference "Bacoloni, A. et al. 2008" was wrongly matched, since the referenced study measured water samples instead of air. | 1. Lines 33- 34, the reference "Bacoloni, A. et al. 2008" has been replaced by "Guo et al., 2016" and "Li et al., 2017".

Lines 535-537: Guo, Z. M., Liu, D., Shen, K.J., Li, J. Yu, Z.Q. Zhang, G.: Concentration and seasonal variation of organophosphorus flame retardants in $PM_{2.5}$ of Taiyuan City, China, Earth and environment (in chinese)., 44, 600-604. https://doi.org/10.14050/j.cnki.1672-9250.2016.06.002, 2016.

Lines 556-558: Li, J., Xie, Z., Mi, W., Lai, S., Tian, C., Emeis, K. C.: Organophosphate esters in air, snow and seawater in the north atlantic and the arctic, Environ. Sci. Technol., 51, 6887-6896. https://doi.org/10.1021/acs.est.7b01289, 2017. |
| 2. Introduction: line 32, the reference "Araki et al. 2014" didn't measured organisms, instead, they measured dust. | 2. Line 35, the reference "Araki et al. 2014" was deleted. |
| 3. Introduction: line 34, the reference "Matthews, et al., 1990, 1993". Both references are animal studies. Thus, stating "many scholars found that OPEs have negative effects on the human body. . ." is not appropriate. | 3. Lines 37-40, the reference "Matthews, et al., 1990, 1993" has been revised to "WHO, 1991, 1998, 2000; Kanazawa et al., 2010; Van der Veen and de Boer, 2012; Du et al., 2015". In addition, "human body" has been revised to "organisms". |
| 4. Introduction: line 41, the reference "Covaci et al. 2007" focused on analytical method development instead of measurement reports, not sure if it is a good reference here. | 4. Line 46-47, the reference "Covaci et al. 2007" has been replaced by "(Möller et al., 2011; 2012; McDonough et al., 2018)". |
| 5. Introduction: line 53, change "14335" to "14,335". | 5. Thank you very much for your advice in such a detail. Line 59, "14335" has been revised to "14,335". |
| 6. Materials and Methods: line 72, (Sigma Aldrich,? location? country?); Be consistent in the text in terms of listing instrument/chemical manufacturing info. | 6. Sigma Aldrich is the reagent production company. The manufacturing information of instruments and reagents has been indicated in Lines 82-88, ("Kelon Chemical Corp., China"; "Sigma |

| | |
|---|---|
| | Aldrich Corp., USA"), and the full text has been checked. |
| 7. Results: line 124, "heavy or light polluted area" may be better. | 7. Line 155, "pollution" has been revised to "polluted". |
| 8. Results: line 126-128, rephrase the sentence to make it more precise. | 8. Lines 155-160, **t**he sentence has been revised to: "These data were quite consistent with our previous study which reported the annual median concentration of OPEs in $PM_{2.5}$ from December 2013 to October 2014 (Yin et al., 2015). Interestingly, the concentration of $\Sigma_7$OPEs at the suburban site was similar to, or even higher than some urban sites, which indicated more local sources of these compounds in the suburban area." |
| 9. Results: line 136, "And they were lower than". | 9. Line 171, it has been revised to: "And they were lower than". |
| 10. Results: line 138, add a space before (Wang, T. et al.), Double check other places in the text to make the format consistent. | 10. Line 174, a space has been add before (Wang, et al.). The typos of the manuscript have been proofread. |
| 11. Results: section 3.3. "Seasonal and spatial variations of OPEs in PM2.5", starting line 186, there is a mis-match in Fig.2 with the context. Where are the seasonal variations presented in Fig.2? Only site variations were presented here. | 11. We are so sorry for this mistake. Because the version we uploaded to the website is different from the first draft, the sequence numbers of figures have been adjusted. We forgot to change it here. Figure 1 refers to "levels and seasonal variation of $\Sigma_7$OPEs at each sampling site". Line 228, "Figure 2" has been revised to "Figure 1". |
| 12. Results: line 227, delete first "the". "Considering" instead of "Considered". | 12. Line 280, first "the" has been deleted. "Considered" has been changed to "Considering". |
| 13. Results: line 228, 229, lowercase "the third ring road". | 13. Line 282, the "third Ring Road" has been revised to "the third ring road". |
| 14. Results: line 229, maybe "the uniform patterns of OPEs levels and distribution across the city is understandable"? | 14. Lines 282-283, this sentence has been revised to "the uniform patterns of OPEs levels and distribution across the city is understandable". |
| 15. Results: line 229, delete "But". | 15. Line 284, "But" has been deleted. |
| 16. Results: line 232, "shoemaking industrial parks are located in the suburbs". | 16. Lines 286-287, this sentence has been revised to "…shoemaking industrial parks are located in the suburbs". |
| 17. Results: line 233, "high levels". | 17. Line 287, "high level" has been revised to "high levels". |

| | |
|---|---|
| 18. Results: line 235, delete "to the individual OPE". | 18. Lines 289-290, "to the individual OPE" has been deleted. |
| 19. Results: line 257, 258, "their gas-particle distributions determine their concentrations in $PM_{2.5}$". | 19. Line 317, "…distributions determines" has been revised to "distributions determine". |
| 20. Results: line 266, is it "Fig.4 showed" or "Fig.5 showed"? | 20. Line 328, "Fig.4 showed" has been revised to "Figure 5 showed". |
| 21. Results: line 275, delete "so". | 21. Line 344, "So" has been deleted. |
| 22. Results: line 282, add "The correlations between" before actually listing pairs of OPE monomers. | 22. Line 351, "The correlations between" has been added. |
| 23. Results: line 284, delete second "was". | 23. Line 353, the second "was" has been deleted. |
| 24. Results: section 3.4.3 "Correlation analysis of OPEs and $PM_{2.5}$ concentrations", you mentioned Fig. S2, in which you used Pearson correlation tests. Why not spearman's rank correlation tests as you used in Figure 5? | 24. As we know, Pearson evaluates the linear relationship between the two variables, while Spearman evaluates the monotonic relationship between the two variables. According to the results of other literature (Wong et al., 2018) and our hypothesis, we think that $PM_{2.5}$ concentration is linearly related to the content of OPEs. So we carried out Pearson correlation tests in Fig. S2 according to the hypothesis. The results showed that the correlation was very poor, which was totally different from what we expected. In order to emphasize the difference of correlation between OPEs/other pollutants and $PM_{2.5}$ concentration, Pearson correlation test result was used.

Lines 138-142, "2.5 Statistical analysis

Data analysis was done through IBM SPSS 22.0. Parameter test and nonparametric test were used to analyze the difference between data. Pearson's correlation coefficients were used to evaluate the linear relationship between the two variables, while Spearman's rank correlation coefficients were used to evaluate the monotonic relationship between the two variables. " has been added. |
| 25. Results: line 291, add "found" after "was". | 25. Line 360, "found" has been added. |
| 26. Results: line 315, "different uses". | 26. Lines 390-391, the sentence has been |

| | revised to "which also indicated the differences of OPEs profile in indoor and outdoor air." |
|---|---|
| 27. Results: line 338,339, add a reference to the statement "Chengdu's wind has always been. . .". | 27. Line 420, a website was added. "(https://baike.baidu.com/item/%E6%88%90%E9%83%BD/128473?fr=aladdin)". "Chengdu is a city located in the interior of China" has been added to illustrate that its wind intensity is smaller than coastal cities. |
| 28. Conclusions and Implications: line 372, "compared to the levels of OPEs in other cities". | 28. Line 457, it has been changed to "Compared to the levels of OPEs in other cities". |
| 29. Conclusions and Implications, line 390, maybe change "not easy to degrade" to "persistent"? What do you mean by "have a high content"?, change the wording to clarify. | 29. Lines 475-476, the sentence has been changed to "The chlorinated phosphate, especially TCPP and TCEP, which are highly toxic and persistent in the environment, have high concentrations in this study." |
| 30. Reference: line 486-488, where the reference was cited? Cannot locate it in the text "Tang, R., Keming, M.A., Zhang, Y., Mao, Q.: Health risk assessment of heavy metals of street dust in Beijing, Acta. Scientiae. Circumstantiae., 32, 2006-2015, https://doi.org/10.13671/j.hjkxxb.2012.08.029, 2012." | 30. Lines 643-645, this reference has been deleted. |
| 31. Reference: what is the novelty in this paper compared with your reference paper in Chinese (Line 512-514) "Yin, H.L., Li, S.P., Ye, Z.X., Yang, Y.C., Liang, J.F., You, J.J.: Pollution Level and Sources of 513 Organic Phosphorus Esters in Airborne PM2.5 in Chengdu City, Environ. Sci. (in chinese), 36, 3566-3572, https://doi.org/10.13227/j.hjkx.2015.10.003, 2015." | 31. The article we published earlier is a report of our experiment results from only two sampling sites. The purpose of that paper was to report the pollution level and distribution of the atmospheric OPEs at urban and suburban sites. Interestingly, we found the seasonal variations of OPEs were significantly different from $PM_{2.5}$ concentrations and $PM_{2.5}$-bound PAHs, etc.. So we carried out a more detailed experiment with six sampling sites in the second year. In this paper, except for reporting the level and seasonal variations of OPEs at six sites, we paid more attention to investigate the relationships and correlations among the target compounds or with influence factors and illustrate the potential sources of OPEs in $PM_{2.5}$. For example, whether different functional areas affect the distributions of |

atmospheric OPEs, correlations of OPEs with environmental factors (temperature, wind, vapor pressure, boiling points, etc.), correlations of OPEs with $PM_{2.5}$ concentrations, correlations of OPEs in $PM_{2.5}$ and soil, correlations of OPEs in indoor and outdoor air were all discussed. These differences are the innovation of this paper.

Lines 66-78: in the revised manuscript, the novelty has been added: "Our previous study has investigated the OPEs concentrations in $PM_{2.5}$ at two sites (urban and suburban sites) in Chengdu (an economically fast growing city in southwest of China), and found that OPEs concentrations and profile were similar at two sites (Yin et al., 2015). But the influence factors and potential sources of OPEs in $PM_{2.5}$ in Chengdu are still unclear. Therefore, in this study, $PM_{2.5}$ was collected over one year (October 2014 to September 2015) at six sites in Chengdu to: a) report the levels and composition profiles of OPEs in urban air in the typical inland city; (b) obtain the seasonal and spatial variation of OPEs in $PM_{2.5}$; (c) investigate the relationships and correlations among the target compounds or with influence factors; (d) illustrate the potential sources of OPEs in $PM_{2.5}$."

| | |
|---|---|
| 32. Reference: line 515-517, reference "Zhang, Q. H., Yang, W. N., Ngo, H. H., Guo, W. S., Jin, P. K., Dzakpasu, M.: Current status of urban wastewater treatment plants in China, Environ. Int., 92-93, 11-22, https://doi.org/10.1016/j.envint.2016.03.024, 2016" might not be a good reference to be used here. | 32. Lines 707-709, this reference has been deleted. |
| 33. Figure 2: where is the seasonal variations? As only site variation is presented here. | 33. There are some errors in the arrangement of the sequence number of the figure. Figure 1 refers to "Levels and seasonal variation of $\Sigma_7$OPEs at each sampling site", and line 205, figure 2 refers to "Percentages of individual OPEs contributing to the $\Sigma_7$OPEs at each sampling site ". Line 228, Figure 2 should actually be Figure 1. |

| | |
|---|---|
| 34. Figure 4: line 542, be consistent with your notations/subscripts in the manuscript, $PM_{2.5}$ or PM2.5. Same issue in line 544 etc. | 34. Thanks for your advice. Line 685, "PM2.5" has been revised to "$PM_{2.5}$". All "PM2.5" appearing in the manuscript has been replaced by "$PM_{2.5}$". |
| 35. Figure 5: Line 544, Should be "Spearman's rank correlation coefficients". Double check other places to be consistent. | 35. Line 336, it has been revised to "Spearman's rank correlation coefficients". We have checked other places throughout the manuscript. |
| 36. Table 1: line 549, "orientation" of what? wind direction? If so, may want to use a different term since suburb and downtown probably do not quite fit. | 36. "Orientation" refers to the direction of the city, not the wind direction.

Line 206, it has been replaced with "Sampling sites". |
| 37. In Figure 5 "Spearman's ranks correlation coefficients between the concentrations of individual OPEs in $PM_{2.5}$ samples" and Figure S2 "Scatter plot of OPEs and $PM_{2.5}$", spearman's rank tests and Pearson's correlation coefficients were used. Could you explain more about the selection of two different correlation tests? | 37. Pearson evaluates the linear relationship between the two variables, while Spearman evaluates the monotonic relationship between the two variables. According to the results of other literatures and our hypothesis, we selected the different test method. In addition, when choosing which of the two test methods to use, firstly the data distribution map was obtained. If it's a normal distribution, Pearson's correlation coefficients were used. If not, Spearman's rank correlation coefficients were used. In the revised manuscript,

Lines 138-142, "2.5 Statistical analysis

Data analysis was done through IBM SPSS 22.0. Parameter test and nonparametric test were used to analyze the difference between data. Pearson's correlation coefficients were used to evaluate the linear relationship between the two variables, while Spearman's rank correlation coefficients were used to evaluate the monotonic relationship between the two variables." was added. |

| Response to reviewer 2: | |
|---|---|
| Questions: | Response: |
| Thanks for the invitation to review. I read the manuscript by Yin et al. with interest. The authors | Dear sir, we are thankful for the reviewer's constructive comments and totally understand the reviewer's concern. The pre-experiment was carried out before the experiment. We conducted the thorough experiment for the quality control and |

reported concentrations of seven OPEs in PM$_{2.5}$ from Chengdu, China, tracked their possible sources, and conducted source apportionment using PCA andthe PMF receptor model. My utmost concern is the data accuracy as some requiredQA/QC procedures were missing. Additionally, the manuscript is a little hard to readas it has a number of grammatical issues, and several statements lacked referencesupports. Though this study provided a few useful information (e.g., difference in OPEprofiles between inland and costal cities), its novelty and quality at current version may not be sufficient enough for the Atmospheric Chemistry and Physics. My specific comments are as follows:

quality assurance including the blank experiment, recoveries of internal standard (TDCPP-d$_{15}$ and TPhP-d$_{15}$) in samples for evaluating the accuracy, blank experiment (field blanks, solvent blanks, matrix blanks), precision experiment, etc.. Due to the limited space of the paper, and the focus of this paper is not on the establishment of analytical methods, it is simplified a lot in the QA/QC part. But we have done all the related experiments for QA/QC, and the results were good. In the revised manuscript, we have added them in QA/QC part. Therefore, there is no need to worry about the accuracy of the data. But it's a pity that there were many grammatical problems and reference problems in the manuscript. We all corrected them and sincerely hope that the manuscript can meet the requirements of Atmospheric Chemistry and Physics after modification. According to your constructive comments, the revisions of the manuscript are as follows:

1. Major concern:
Novelty: There is a similar study previously conducted by the leading author here. What makes this manuscript distinct from that previous one? Authors should elaborate more the novelty of this study.

The article we published earlier is a report of our experiment results from only two sampling sites. The purpose of that paper was to report the pollution level and distribution of the atmospheric OPEs at urban and suburban sites. Interestingly, we found the seasonal variations of OPEs were significantly different from PM$_{2.5}$ concentrations and PM$_{2.5}$-bound PAHs, etc.. So we carried out a more detailed experiment with six sampling sites in the second year. In this paper, except for reporting the level and seasonal variations of OPEs at six sites, we paid more attention to investigate the relationships and correlations among the target compounds or with influence factors and illustrate the

| | potential sources of OPEs in PM$_{2.5}$. For example, whether different functional areas affect the distributions of atmospheric OPEs, correlations of OPEs with environmental factors (temperature, wind, vapor pressure, boiling points, etc.), correlations of OPEs with PM$_{2.5}$ concentrations, correlations of OPEs in PM$_{2.5}$ and soil, correlations of OPEs in indoor and outdoor air were all discussed. These differences are the innovation of this paper. |
|---|---|
| | In the revised manuscript, the novelty has been added in lines 66-78: "Our previous study has investigated the OPEs concentrations in PM$_{2.5}$ at two sites (urban and suburban sites) in Chengdu (an economically fast growing city in southwest of China), and found that OPEs concentrations and profile were similar at two sites (Yin et al., 2015). But the influence factors and potential sources of OPEs in PM$_{2.5}$ in Chengdu are still unclear. Therefore, in this study, PM$_{2.5}$ was collected over one year (October 2014 to September 2015) at six sites in Chengdu to: a) report the levels and composition profiles of OPEs in urban air in the typical inland city; (b) obtain the seasonal and spatial variation of OPEs in PM$_{2.5}$; (c) investigate the relationships and correlations among the target compounds or with influence factors; (d) illustrate the potential sources of OPEs in PM$_{2.5}$." |
| QA/QC: 1) As no surrogate standards were spiked prior to sample treatment, how did authors evaluate OPE recoveries from the analytical procedures? | Lines 127-132, "Thorough QA/QC procedures for OPE analysis were conducted to ensure data quality. To evaluate the recovery efficiencies of analytical procedures, all samples were added with internal standard (TDCPP-d$_{15}$ and TPhP-d$_{15}$), and the accuracy was evaluated by their recoveries. The concentrations of the 7 OPEs were determined by an external standard method. The correlation coefficients of the standard curves of the seven OPE monomers were all greater than 0.990. The recoveries of 7 OPEs and the internal standard were between 78.9% and 122.5%." was added. |
| | Lines 135-137, "Field blanks were done at each site to evaluate the background contamination in the field. TBEP, TnBP and TEHP were detected in it. The level of them found in the blank were <15% of the concentrations measured in all samples." was added. |
| | There are only two internal standards, so we use them to ensure the recovery, but use external standard method to quantify the target compounds. In addition, a matrix blank was run in |

| | parallel with every batch of samples for the analysis of OPEs. Only TnBP was detected in the blanks, and the level of TnBP found in the blanks was <5% of the concentrations measured in all samples, which meant it was negligible. The correlation coefficients of the standard curves of the seven OPE profiles were all greater than 0.990. These all could ensure the accuracy of the data. |
|---|---|
| 2) How was the matrix effect assessed and compensated? | The matrix effect was assessed by the matrix blank experiment. The blank quartz membrane was added with the internal standard (TDCPP-d$_{15}$ and TPhP-d$_{15}$) and OPEs standard. After the whole pretreatment process, the recoveries of 7 OPEs and the internal standard were all between 70% and 120%. So the data was not corrected and the matrix effect was not compensated. |
| 3) The data from field blanks were missing. PMF model: How were the uncertainties determined? Which references were referred to for identification of sources associated with each factor? I also want to see the source profile of each factor. | The field blanks were done which were prepared and installed in the same manner as the regular samples but without turning on the sampler motor. Due to the limited space of the paper, and the focus of this paper is not on the establishment of analytical methods, so it is simplified a lot in the QA/QC part. But we have done all the related experiments for QA/QC, and the results were good. |

Lines 135-137, "Field blanks were done at each site to evaluate the background contamination in the field." TBEP, TnBP and TEHP were detected in it. The level of them found in the blank were <15% of the concentrations measured in all samples." was added.

Lines 445-446, "The uncertainty is estimated by three methods: BS, disp and bs-disp" was added for PMF. The results are shown in the table below:

DISP results showed that the solution was stable because no swaps were present.

BS results showed that mapping over 80% of the factors indicated that the BS uncertainties could be interpreted and the number of factors may be appropriate.

All of the "Strong" species were selected for the BS-DISP error estimation. The number of DISP and BS-DISP swaps was zero. BS-DISP highlight that the solution may be reliable due to there was no swaps across two factors.

Error estimation summary results

| **BS-DISP Diagnostics:** | | | |
|---|---|---|---|

| | | | |
|---|---|---|---|
| # of Cases Accepted: | 100 | | |
| % of Cases Accepted: | 100% | | |
| Largest Decrease in Q: | -0.150999993 | | |
| %dQ: | -0.067824623 | | |
| # of Decreases in Q: | 0 | | |
| # of Swaps in Best Fit: | 0 | | |
| # of Swaps in DISP: | 0 | | |
| Swaps by Factor: | 0 | 0 | 0 |
| **DISP Diagnostics:** | | | |
| Error Code: | 0 | | |
| Largest Decrease in Q: | -0.005 | | |
| %dQ: | -0.002245848 | | |
| Swaps by Factor: | 0 | 0 | 0 |

| **BS Mapping:** | | | | |
|---|---|---|---|---|
| | Factor 1 | Factor 2 | Factor 3 | Unmapped |
| Boot Factor 1 | 100 | 0 | 0 | 0 |
| Boot Factor 2 | 0 | 100 | 0 | 0 |
| Boot Factor 3 | 0 | 0 | 100 | 0 |

The source profile of each factor:

| Factor Profiles (% of species sum) from Base Run (Convergent Run) | | | | | | | |
|---|---|---|---|---|---|---|---|
| | TnBP | TCEP | TCPP | TDCPP | TPhP | TBEP | TEHP |
| Factor 1 | 28.69 | 70.95 | 70.72 | 31.01 | 58.34 | 0.00 | 70.93 |
| Factor 2 | 0.00 | 20.31 | 25.47 | 44.72 | 13.97 | 77.95 | 26.41 |
| Factor 3 | 71.31 | 8.73 | 3.81 | 24.27 | 27.69 | 22.05 | 2.66 |

Lines 450-455, the references were referred to for identification of sources associated with each factor. "Factor 1 was deduced to be the plastics/electrical industry and indoor source emissions (Esch, 2000; Stevens et al., 2006). Factor 2 contributed the most to TBEP (78.0%), followed by TDCPP (44.7%), while it did not contribute to TnBP. Therefore, factor 2 was deduced as the food/cosmetics industry and traffic emissions (Marklund et al., 2005). Factor 3 contributes 71.7% of the total TnBP, which can be deduced as chemical industrial source (Regnery et al., 2011)."

| | |
|---|---|
| Minor concern:
Line 8: "emerging contaminants"→"contaminant of emerging concern". OPEs have been produced for decades. | Line 10, "emerging contaminants" has been revised to "OPEs are contaminants of emerging concern". |

| | |
|---|---|
| Line 9: "centers"→ "areas" | Line 11, "Centers" has been replaced by "areas". |
| Line 13... which TOGETHER made up..." | Line 15, "which made up" has been revised to "which together made up". |
| Line 18: OPEs can transfer from soil to air particles via suspension and volatilization as well. Actually, authors mentioned this at Lines 303-304. | Line 20, "suggested the atmospheric $PM_{2.5}$ settlement is an important source of OPEs in soil" has been deleted. The sentence has been revised to "Very strong correlation ($R^2 = 0.98$, p<0.01) between the OPEs in soil and in $PM_{2.5}$ was observed." |
| Lines 32-35: A weird sentence, please rephrase it. | Lines 36-38, the sentence has been revised to "However, many scholars found that the residues of OPEs in the environment could cause toxic effects on organisms (WHO, 1991, 1998, 2000; Kanazawa et al., 2010; Van der Veen and de Boer, 2012; Du et al., 2015)". |
| Line 35: Reference is needed for the "OPE restrictions". | Lines 40-41, three references were added "Some countries have legislated to restrict the usage of OPEs (Blum et al., 2019; Exponent, 2018; State of California, 2020)". Lines 495-498, "Blum, A.; Behl, M.; Birnbaum, L. S.; Diamond, M. L.; Phillips, A.; Singla, V.; Sipes, N. S.; Stapleton, H. M.; Venier, M. Organophosphate ester flame retardants: Are they a regrettable substitution for polybrominated diphenyl ethers? Environ. Sci. Technol. Lett. 2019, 6, 638-649. " was added. Lines 532-534, "Exponent. California bans flame retardants in certain consumer products. 2018, Available at: https://www.exponent.com/knowledge/alerts/2018/09/california-bans-flame-retardants/?pageSize=NaN&pageNum=0&loadAllByPageSize=true (accessed February 15, 2020). State of California. Safer consumer products (SCP) information management system. 2020. Available at: https://calsafer.dtsc.ca.gov/cms/search/?type=Chemical (accessed February 21, 2020)." were added. |
| Lines 38-39: Reference is needed. | Lines 44-47, references have been added: "The detection of OPEs in Arctic and Antarctic snow samples and atmospheric particulate matter samples demonstrated that OPEs can be transported over long distances (Möller et al., 2012; Li et al., 2017). Studies on OPEs in oceans were carried out a lot, and the concentrations of particle-bound OPEs ranged from tens to |

thousands of ng m$^{-3}$ (Möller et al., 2011; 2012; Cristale J & Lacorte S., 2013; Li et al., 2017; McDonough et al., 2018)".

Lines 583-592, "McDonough, C. A., De Silva, A. O., Sun, C., Cabrerizo, A., Adelman, D., Soltwedel, T., Bauerfeind, E., Muir, D. C. G., Lohmann, R.: Dissolved organophosphate esters and polybrominated diphenyl ethers in remote marine environments: Arctic surface water distributions and net transport through Fram Strait, Environ. Sci. Technol., 52, 6208-6216, https://doi.org/10.1021/acs.est.8b01127, 2018.

Möller, A.; Sturm, R.; Xie, Z.; Cai, M.; He, J.; Ebinghaus, R. Organophosphorus flame retardants and plasticizers in airborne particles over the Northern Pacific and Indian Ocean toward the polar regions: Evidence for global occurrence. Environ. Sci. Technol. 2012, 46, 3127-3134.

Möller, A.; Xie, Z.; Caba, A.; Sturm, R.; Ebinghaus, R. Organophosphorus flame retardants and plasticizers in the atmosphere of the North Sea. Environ. Pollut. 2011, 159, 3660-3665. https://doi.org/10.1016/j.envpol.2011.07.022, 2011." have been added.

| Line 45: Which type of matrix is referred to for "Concentrations of OPEs in most cities..."I looked at the references cited, but not all of them talked about PM$_{2.5}$. | This matrix is only for outdoor atmospheric environment.

Line 51, "Concentrations of OPEs in most cities..." has been revised to "Concentrations of atmospheric OPEs in most cities". Not all of the references we cited talked about PM$_{2.5}$, but they were all about OPEs in atmospheric particles. |
|---|---|
| Lines 54-56: How about "Chengdu is an important city in Southwest China due to its role as a national high-tech industrial base, a commercial logistics center, and a comprehensive transportation hub"? | Line 62, a website has been added: "(https://en.wikipedia.org/wiki/Chengdu)." |
| Line82: Sampling intervals? | Lines 94-100, "In each season, continuous sampling was carried out for about one week, except for rainy days. In autumn, the sampling duration was from October 23 to October 29, 2014 (no sample was obtained due to the rain on October 26 and 27); in winter, the sampling duration was from December 22 to December 30, 2014 (no sample was obtained due to the rain on October 25 and 26); in spring, the sampling duration was from March 25 to March 30, 2015; in summer, the sampling duration |

| | was from July 16 to July 24, 2015 (no sample was obtained due to the rain on July 21)" has been added. Each collection campaign lasted 23 h. The interval of each sample was 1h. |
|---|---|
| Line 86: Was the analytical method used here applied in any previous studies? | Based on the references of Möller et al. (2012), we established the quantitative analysis method in the laboratory. The analytical method has been applied in our previous studies. Li, S. P., Yin, H. L., Ye, Z. X., Liang, J. F., Hao, Y. F.: GC-MS determination of 7 organic phosphate ester flame retardants in atmospheric particulates with chromatography purification: PTCA (Part B: Chem Anal). 2015, 051(005), 581-585, https://doi.org/10.13227/j.hjkx.2015.10.003, 2015. Yin, H.L., Li, S.P., Ye, Z.X., Yang, Y.C., Liang, J.F., You, J.J.: Pollution level and sources of organic phosphorus esters in airborne $PM_{2.5}$ in Chengdu City, Environ. Sci. (in chinese), 36, 3566-3572, https://doi.org/10.13227/j.hjkx.2015.10.003, 2015. Möller, A.; Sturm, R.; Xie, Z.; Cai, M.; He, J.; Ebinghaus, R. Organophosphorus flame retardants and plasticizers in airborne particles over the Northern Pacific and Indian Ocean toward the polar regions: Evidence for global occurrence. Environ. Sci. Technol. 2012, 46, 3127-3134. |
| Lines 93-94: How about "The latter eluate was collected and con-centrated by vacuum-condensing..."? | The latter eluate was collected in a centrifugal tube and then concentrated to nearly dry by vacuum condensing equipment and then fixed volume to 200 μL with hexane. Then it was placed in a sample bottle to wait for the injection of gas chromatography-mass spectrometry (GC-MS). Lines 111-113, it has been revised to "...the latter eluate (ethyl acetate/acetone) was collected. The eluate was concentrated to nearly dry by vacuum-condensing equipment and then fixed volume to 200 $\mu$L with hexane for gas chromatography-mass spectrometry (GC-MS) (Shimadzu 2010plus, Japan) analysis." in the manuscript. |
| Lines 114-118: Could concisely say "detected in virtually all the samples". | Lines 145-148, "Four OPEs (TCPP, TDCPP, TCEP and TnBP) were detected in all samples (n=149), while TBEP was detected in all but one sample. Additionally, TEHP was detected in 96.7% of samples overall and TPhP was detected in 98% of samples." has been revised to "Seven OPEs were detected in 96.7% - 100% of the samples (n=149)". |
| Lines 120-121: Did "The average value... four seasons mean "annual average level"? | It means "seasonal average concentration", not "annual average level". Lines 151-153, it has been revised to "The seasonal average value of OPEs in $PM_{2.5}$ at each site was almost |

| | at the same level (5.8 $\pm$ 1.3 ng m$^{-3}$-6.9 $\pm$ 2.5 ng m$^{-3}$)". We have checked other places throughout the manuscript. |
|---|---|
| Line 141: Rephrase the first sentence. | Line 176, it has been revised to "Non-chlorinated OPEs were the predominant OPEs across Chengdu city". |
| Lines 143-145: Explain the meaning of values in the parentheses. | Line 179, it has been revised to "(annual media concentration: 2.3 ng m$^{-3}$, 35.3% of $\Sigma_7$ OPEs)". |
| Lines: 165-167: References? | Line 202, two company websites for producing and selling OPEs have been added: "https://show.guidechem.com/hainuowei, http://www.sinostandards.net/index.php". |
| Lines182-184: A recent study measuring an extended list of OPEs in the Great Lakes atmosphere also found that alkyl OPEs dominated OPE compositional profiles of urban air collected from Chicago and Cleveland (Wu et al. 2020; 10.1021/acs. est.9b07755). | Thank you for your reminder. We have referred to the results of this study. For example, (1) lines 220-221, "Wu et al. (2020) also reported that alkyl OPEs dominated OPE compositional profiles of urban air collected from Chicago and Cleveland." (2) Lines 247-252, "Wu et al. (2020) reported that median concentrations of $\Sigma$ OPEs for summer samples were up to 5 times greater than those for winter samples. The similar seasonal patterns were reported by Salamova et al. (2014) for the atmospheric particle-phase OPE concentrations in samples collected from the Great Lakes in 2012. A reasonable explanation is that OPEs are not chemically bound to the materials in which they are used and higher temperatures may facilitate their emission from buildings and vehicles." has been added. (3) Lines 290-293, "Interestingly, in this study, alkyl OPEs dominated both urban and suburban sites. This was extremely different from the results reported by Wu et al. (2020) that alkyl OPEs dominated at urban sites, chlorinated OPEs were prevalent at rural sites, and aryl OPEs were most abundant at remote locations." has been added in the revised manuscript. |
| Line 208-210: OPE levels can be surely affected by temperature, so I suppose the authors would like to say "seasonal variations in OPE levels". Additionally, would meteorological parameters other than temperature result in the seasonal variations found in the present study? | Based on our experience, we also strongly agree that temperature and other meteorological factors will affect the level of pollutants in PM$_{2.5}$. However, the concentration of OPEs found in this study did not varied much in the four seasons, which was significantly different from other pollutants. Some literatures showed that the seasonal variations of OPEs in some coastal cities were significantly affected by temperature (Liu et al., 2016; Wang et al., 2019). For example, Wang et al. (2019) reported seasonality was discovered for OPEs in both gas phase and PM$_{2.5}$ with their concentrations higher in hot seasons in |

| | |
|---|---|
| | Dalian, which may due to the temperature-driven emission or gas-particle partitioning. |
| | Lines 254-259, it has been revised to "In our study, the correlation analysis between the temperature, wind speed, wind direction and $\Sigma_7$OPEs concentrations has been done. The results showed statistically significant negative correlations between temperature and $\Sigma_7$ OPEs (R= -0.355, p<0.01). The lowest concentrations of $\Sigma_7$OPEs and individual compound were observed in summer suggesting the OPEs level was not driven by the temperature-driven emission. Gas-particle partitioning and local emission sources may contribute to the variation." In addition, other meteorological parameters with high contributions to the seasonal variations were not found in the present study. |
| Lines 236-238: Has been mentioned before. Lines 238-248: Out of place here. Could be moved to section 3.1. | Lines 295-297 have been deleted. |
| Line 257: Need reference to support "they tend to be adsorbed in PM $_{2.5}$". | Line 316, reference has been added: " (Wang et al., 2019)". Lines 675-677, "Wang, Y., Bao. M. J., Tan. F., Qu. Z. P., Zhang. Y. W., Chen. J. W.: Distribution of organophosphate esters between the gas phase and $PM_{2.5}$ in urban Dalian, China, Environ. Pollut., https://doi.org/10.1016/j.envpol.2019.113882, 2019." |
| Line 315: Other factors may lead to such difference between indoor and outdoor OPEs. For example, TBEP has the shortest atmospheric half-lives, which may explain why its dominance in indoor samples was not observed for the outdoor counterparts. | Of course, other factors may also cause differences in the content of indoor and outdoor OPEs. Lines 385-395 have been revised to "Except for the different usage of OPEs, many factors may also lead to differences between indoor and outdoor OPEs. For example, TBEP has the shortest atmospheric half-lives, which may explain why its dominance in indoor samples was not observed for the outdoor counterparts. Studies in Swedish (Wong et al., 2018) reported the concentrations of OPEs in indoor air were TCPP > TCEP > TBEP > TnBP> TPhP, and in outdoor urban air were TBEP > TCPP > TCEP > TnBP > TPhP which also indicated the differences of OPEs profile in indoor and outdoor air. They found that activities in the building, e.g. floor cleaning, polishing, construction, introduction of new electronics and changes in ventilation rate could be key factors in controlling the concentration of indoor air pollutants, while the observed seasonality for OPEs in outdoor air was due to changes in |

| | primary emission." |
|---|---|
| Lines 350-356: References are required for identification of possible sources associated with each factor. | Thanks for your advice. Lines 433-440, references have been added: "Factor 1 can represent the sources of OPEs from the plastic industry, interior decoration and traffic emission, with the contribution ratio of 34.5% (Marklund et al., 2005; Regnery et al., 2011; CEFIC, 2002). Factor 2 has higher load on TnBP, TEHP and TPhP. The highest load was on TnBP, which is often used as a high-carbon alcohol defoamer, mostly in industries that do not come in contact with food and cosmetics, as well as in antistatic agents and extractants of rare earth elements. TEHP can be used as an antifoaming agent, hydraulic fluid and so on. TPhP is typically used in electrical and electronic products, or plastic film and rubber (Esch, 2000; Stevens et al., 2006; Wei et al., 2015)." |

 **Response to Anonymous Referee #1**

Interactive comment on "Measurement report: Seasonal, distribution and sources of organophosphate esters in PM$_{2.5}$ from an inland urban city in southwest China" by Hongling Yin et al.

Anonymous Referee #1

Reviews: The manuscript reported the measurement of OPEs in PM$_{2.5}$ in Chengdu, China and presented the seasonal and spatial distributions, and the potential sources of the OPEs by using multiple correlation tests. The analysis and reported data were consistent with the conclusions. The measurements and findings are critical to fill in the knowledge gap of OPEs levels in inland cities.

However, several issues need to be addressed before acceptance for publication. Besides some typos and wording changes, Figure 2 seems not matching the context since no seasonal variations can be seen.

Since different statistical tests were used, e.g. Pearson correlation test, spearC1 ACPD Interactive comment Printer-friendly version Discussion paper man's rank correlation test, and nonparametric test, a clearer statement of conditions (e.g. normality check) to use those tests is needed. Lastly, the references need to be checked carefully since some of them are either not matched or not cited appropriately.

**Response:** Thank you for your valuable comments and good advice on improving our manuscript.

We are so sorry that the manuscript has some mistakes. The typos and wording of the manuscript, as well as the specific contents and references of the manuscript, have been revised as follows according to your comments.

Specific comments on the manuscript

1. Introduction: line 30, the reference "Bacoloni, A. et al. 2008" was wrongly matched, since the referenced study measured water samples instead of air.

**Response:** Lines 33- 34, the reference "Bacoloni, A. et al. 2008" has been replaced by "Guo et al.,

2016" and "Li et al., 2017".

Lines 535-537: Guo, Z. M., Liu, D., Shen, K.J., Li, J. Yu, Z.Q. Zhang, G.: Concentration and seasonal variation of organophosphorus flame retardants in PM$_{2.5}$ of Taiyuan City, China, Earth and environment (in chinese)., 44, 600-604. https://doi.org/10.14050/j.cnki.1672-9250.2016.06.002, 2016.

Lines 556-558: Li, J., Xie, Z., Mi, W., Lai, S., Tian, C., Emeis, K. C.: Organophosphate esters in air, snow and seawater in the north atlantic and the arctic, Environ. Sci. Technol., 51, 6887-6896.

https://doi.org/10.1021/acs.est.7b01289, 2017.

2. Introduction: line 32, the reference "Araki et al. 2014" didn't measured organisms, instead, they measured dust.

**Response:** Line 35, the reference "Araki et al. 2014" was deleted.

3. Introduction: line 34, the reference "Matthews, et al., 1990, 1993". Both references are animal studies. Thus, stating "many scholars found that OPEs have negative effects on the human body. . ." is not appropriate.

**Response:** Lines 37-40, the reference "Matthews, et al., 1990, 1993" has been revised to "WHO,

1991, 1998, 2000; Kanazawa et al., 2010; Van der Veen and de Boer, 2012; Du et al., 2015". In addition, "human body" has been revised to "organisms".

4. Introduction: line 41, the reference "Covaci et al. 2007" focused on analytical method development instead of measurement reports, not sure if it is a good reference here.

**Response:** Line 46-47, the reference "Covaci et al. 2007" has been replaced by "(Möller et al.,

2011; 2012; McDonough et al., 2018)".

5. Introduction: line 53, change "14335" to "14,335".

**Response:** Thank you very much for your advice in such a detail. Line 59, "14335" has been revised to "14,335".

6. Materials and Methods: line 72, (Sigma Aldrich,? location? country?); Be consistent in the text in terms of listing instrument/chemical manufacturing info.

**Response:** Sigma Aldrich is the reagent production company. The manufacturing information of instruments and reagents has been indicated in Lines 82-88, ("Kelon Chemical Corp., China"; "Sigma

Aldrich Corp., USA"), and the full text has been checked.

7. Results: line 124, "heavy or light polluted area" may be better.

**Response:** Thanks for your advice. Line 155, "pollution" has been revised to "polluted".

8. Results: line 126-128, rephrase the sentence to make it more precise.

**Response:** Lines 155-160, **t**he sentence has been revised to: "These data were quite consistent with our previous study which reported the annual median concentration of OPEs in $PM_{2.5}$ from

December 2013 to October 2014 (Yin et al., 2015). Interestingly, the concentration of $\Sigma_7$OPEs at the suburban site was similar to, or even higher than some urban sites, which indicated more local sources of these compounds in the suburban area."

9. Results: line 136, "And they were lower than".

**Response:** line 171, it has been revised to: "And they were lower than". The grammatical problems in this manuscript have been carefully corrected.

10. Results: line 138, add a space before (Wang, T. et al.), Double check other places in the text to make the format consistent.

**Response:** Line 174, a space has been add before (Wang, et al.). The typos of the manuscript have been proofread.

11. Results: section 3.3. "Seasonal and spatial variations of OPEs in PM2.5", starting line 186, there is a mis-match in Fig.2 with the context. Where are the seasonal variations presented in Fig.2?

Only site variations were presented here.

**Response:** We are so sorry for this mistake. Because the version we uploaded to the website is different from the first draft, the sequence numbers of figures have been adjusted. We forgot to change it here. Figure 1 refers to "levels and seasonal variation of $\Sigma_7$OPEs at each sampling site". Line 228,

"Figure 2" has been revised to "Figure 1".

12. Results: line 227, delete first "the". "Considering" instead of "Considered".

**Response:** Line 280, first "the" has been deleted. "Considered" has been changed to

"Considering".

13. Results: line 228, 229, lowercase "the third ring road".

**Response:** Line 282, the "third Ring Road" has been revised to "the third ring road".

14. Results: line 229, maybe "the uniform patterns of OPEs levels and distribution across the city is understandable"?

**Response:** Thanks a lot. Lines 282-283, this sentence has been revised to "the uniform patterns of

OPEs levels and distribution across the city is understandable".

15. Results: line 229, delete "But".

**Response:** Line 284, "But" has been deleted.

16. Results: line 232, "shoemaking industrial parks are located in the suburbs".

**Response:** Lines 286-287, this sentence has been revised to "…shoemaking industrial parks are located in the suburbs". Thank you for your advice.

833  17. Results: line 233, "high levels".

834  **Response:** Line 287, "high level" has been revised to "high levels".

835  18. Results: line 235, delete "to the individual OPE".

836  **Response:** Lines 289-290, "to the individual OPE" has been deleted.

837  19. Results: line 257, 258, "their gas-particle distributions determine their concentrations in

838 $PM_{2.5}$".

839  **Response:** Line 317, "…distributions determines" has been revised to "distributions determine".

840  20. Results: line 266, is it "Fig.4 showed" or "Fig.5 showed"?

841  **Response:** Line 328, "Fig.4 showed" has been revised to "Figure 5 showed".

842  21. Results: line 275, delete "so".

843  **Response:** Line 344, "So" has been deleted. Thanks for your advice.

844  22. Results: line 282, add "The correlations between" before actually listing pairs of OPE

845 monomers.

846  **Response:** Line 351, "The correlations between" has been added.

847  23. Results: line 284, delete second "was".

848  **Response:** Line 353, the second "was" has been deleted.

849  24. Results: section 3.4.3 "Correlation analysis of OPEs and $PM_{2.5}$ concentrations", you

850 mentioned Fig. S2, in which you used Pearson correlation tests. Why not spearman's rank correlation

851 tests as you used in Figure 5?

852  **Response:** As we know, Pearson evaluates the linear relationship between the two variables,

853 while Spearman evaluates the monotonic relationship between the two variables. According to the

854 results of other literature (Wong et al., 2018) and our hypothesis, we think that $PM_{2.5}$ concentration is

855 linearly related to the content of OPEs. So we carried out Pearson correlation tests in Fig. S2 according

856 to the hypothesis. The results showed that the correlation was very poor, which was totally different

857 from what we expected. In order to emphasize the difference of correlation between OPEs/other

858 pollutants and $PM_{2.5}$ concentration, Pearson correlation test result was used.

859  Line 138-142, "2.5 Statistical analysis

860  Data analysis was done through IBM SPSS 22.0. Parameter test and nonparametric test were used

861 to analyze the difference between data. Pearson's correlation coefficients were used to evaluate the linear relationship between the two variables, while Spearman's rank correlation coefficients were used to evaluate the monotonic relationship between the two variables. " has been added.

25. Results: line 291, add "found" after "was".

**Response:** Line 360, "found" has been added.

26. Results: line 315, "different uses".

**Response:** Lines 390-391, the sentence has been revised to "which also indicated the differences of OPEs profile in indoor and outdoor air."

27. Results: line 338,339, add a reference to the statement "Chengdu's wind has always been. . .".

**Response:**    Line    420,    a    website    was    added.

"(https://baike.baidu.com/item/%E6%88%90%E9%83%BD/128473?fr=aladdin)". "Chengdu is a city located in the interior of China" has been added to illustrate that its wind intensity is smaller than coastal cities.

28. Conclusions and Implications: line 372, "compared to the levels of OPEs in other cities".

**Response:** Line 457, it has been changed to "Compared to the levels of OPEs in other cities".

29. Conclusions and Implications, line 390, maybe change "not easy to degrade" to "persistent"?

What do you mean by "have a high content"?, change the wording to clarify.

**Response:** Lines 475-476, the sentence has been changed to "The chlorinated phosphate, especially TCPP and TCEP, which are highly toxic and persistent in the environment, have high concentrations in this study."

30. Reference: line 486-488, where the reference was cited? Cannot locate it in the text "Tang, R.,

Keming, M.A., Zhang, Y., Mao, Q.: Health risk assessment of heavy metals of street dust in Beijing,

Acta. Scientiae. Circumstantiae., 32, 2006-2015, https://doi.org/10.13671/j.hjkxxb.2012.08.029, 2012."

**Response:** Lines 643-645, this reference has been deleted.

31. Reference: what is the novelty in this paper compared with your reference paper in Chinese (Line 512-514) "Yin, H.L., Li, S.P., Ye, Z.X., Yang, Y.C., Liang, J.F., You, J.J.: Pollution Level and

Sources of 513 Organic Phosphorus Esters in Airborne PM2.5 in Chengdu City, Environ. Sci. (in chinese), 36, 3566-3572, https://doi.org/10.13227/j.hjkx.2015.10.003, 2015."

**Response:** The article we published earlier is a report of our experiment results from only two sampling sites. The purpose of that paper was to report the pollution level and distribution of the atmospheric OPEs at urban and suburban sites. Interestingly, we found the seasonal variations of OPEs were significantly different from $PM_{2.5}$ concentrations and $PM_{2.5}$-bound PAHs, etc.. So we carried out a more detailed experiment with six sampling sites in the second year. In this paper, except for reporting the level and seasonal variations of OPEs at six sites, we paid more attention to investigate the relationships and correlations among the target compounds or with influence factors and illustrate the potential sources of OPEs in $PM_{2.5}$. For example, whether different functional areas affect the distributions of atmospheric OPEs, correlations of OPEs with environmental factors (temperature, wind, vapor pressure, boiling points, etc.), correlations of OPEs with $PM_{2.5}$ concentrations, correlations of

OPEs in $PM_{2.5}$ and soil, correlations of OPEs in indoor and outdoor air were all discussed. These differences are the innovation of this paper.

Lines 66-78: in the revised manuscript, the novelty has been added: "Our previous study has investigated the OPEs concentrations in $PM_{2.5}$ at two sites (urban and suburban sites) in Chengdu (an economically fast growing city in southwest of China), and found that OPEs concentrations and profile were similar at two sites (Yin et al., 2015). But the influence factors and potential sources of OPEs in

$PM_{2.5}$ in Chengdu are still unclear. Therefore, in this study, $PM_{2.5}$ was collected over one year (October

2014 to September 2015) at six sites in Chengdu to: a) report the levels and composition profiles of

OPEs in urban air in the typical inland city; (b) obtain the seasonal and spatial variation of OPEs in

$PM_{2.5}$; (c) investigate the relationships and correlations among the target compounds or with influence factors; (d) illustrate the potential sources of OPEs in $PM_{2.5}$."

32. Reference: line 515-517, reference "Zhang, Q. H., Yang, W. N., Ngo, H. H., Guo, W. S., Jin, P.

K., Dzakpasu, M.: Current status of urban wastewater treatment plants in China, Environ. Int., 92-93,

11-22, https://doi.org/10.1016/j.envint.2016.03.024, 2016" might not be a good reference to be used here.

**Response:** Lines 707-709, this reference has been deleted.

33. Figure 2: where is the seasonal variations? As only site variation is presented here.

**Response:** There are some errors in the arrangement of the sequence number of the figure. Figure

1 refers to "Levels and seasonal variation of $\Sigma_7$OPEs at each sampling site", and line 205, figure 2

refers to "Percentages of individual OPEs contributing to the $\Sigma_7$OPEs at each sampling site ". Line 228,

Figure 2 should actually be Figure 1.

34. Figure 4: line 542, be consistent with your notations/subscripts in the manuscript, $PM_{2.5}$ or

PM2.5. Same issue in line 544 etc.

**Response:** Thanks for your advice. Line 685, "PM2.5" has been revised to "$PM_{2.5}$". All "PM2.5"

appearing in the manuscript has been replaced by "$PM_{2.5}$".

35. Figure 5: Line 544, Should be "Spearman's rank correlation coefficients". Double check other places to be consistent.

**Response:** Line 336, it has been revised to "Spearman's rank correlation coefficients". We have checked other places throughout the manuscript.

36. Table 1: line 549, "orientation" of what? wind direction? If so, may want to use a different term since suburb and downtown probably do not quite fit.

**Response:** "Orientation" refers to the direction of the city, not the wind direction.

Line 206, it has been replaced with "Sampling sites".

37. In Figure 5 "Spearman's ranks correlation coefficients between the concentrations of individual OPEs in $PM_{2.5}$ samples" and Figure S2 "Scatter plot of OPEs and $PM_{2.5}$", spearman's rank tests and Pearson's correlation coefficients were used. Could you explain more about the selection of two different correlation tests?

**Response:** Pearson evaluates the linear relationship between the two variables, while Spearman evaluates the monotonic relationship between the two variables. According to the results of other literatures and our hypothesis, we selected the different test method. In addition, when choosing which of the two test methods to use, firstly the data distribution map was obtained. If it's a normal distribution, Pearson's correlation coefficients were used. If not, Spearman's rank correlation coefficients were used. In the revised manuscript,

Lines 138-142, "2.5 Statistical analysis

Data analysis was done through IBM SPSS 22.0. Parameter test and nonparametric test were used to analyze the difference between data. Pearson's correlation coefficients were used to evaluate the linear relationship between the two variables, while Spearman's rank correlation coefficients were used to evaluate the monotonic relationship between the two variables." was added.

 **Response to Anonymous Referee #2**

Thanks for the invitation to review. I read the manuscript by Yin et al. with interest. The authors reported concentrations of seven OPEs in PM$_{2.5}$ from Chengdu, China,tracked their possible sources, and conducted source apportionment using PCA andthe PMF receptor model. My utmost concern is the data accuracy as some requiredQA/QC procedures were missing. Additionally, the manuscript is a little hard to readas it has a number of grammatical issues, and several statements lacked referencesupports.

Though this study provided a few useful information (e.g., difference in OPEprofiles between inland and costal cities), its novelty and quality at current version may not be sufficient enough for the

Atmospheric Chemistry and Physics. My specific comments are as follows:

**Response:** Dear sir, we are thankful for the reviewer's constructive comments and totally understand the reviewer's concern. The pre-experiment was carried out before the experiment. We conducted the thorough experiment for the quality control and quality assurance including the blank experiment, recoveries of internal standard (TDCPP-d$_{15}$ and TPhP-d$_{15}$) in samples for evaluating the accuracy, blank experiment (field blanks, solvent blanks, matrix blanks), precision experiment, etc..

Due to the limited space of the paper, and the focus of this paper is not on the establishment of analytical methods, it is simplified a lot in the QA/QC part. But we have done all the related experiments for QA/QC, and the results were good. In the revised manuscript, we have added them in

QA/QC part. Therefore, there is no need to worry about the accuracy of the data. But it's a pity that there were many grammatical problems and reference problems in the manuscript. We all corrected them and sincerely hope that the manuscript can meet the requirements of Atmospheric Chemistry and

Physics after modification. According to your constructive comments, the revisions of the manuscript are as follows:

Major concern:

Novelty: There is a similar study previously conducted by the leading author here. What makes this manuscript distinct from that previous one? Authors should elaborate more the novelty of this study.

**Response:** The article we published earlier is a report of our experiment results from only two sampling sites. The purpose of that paper was to report the pollution level and distribution of the atmospheric OPEs at urban and suburban sites. Interestingly, we found the seasonal variations of OPEs were significantly different from $PM_{2.5}$ concentrations and $PM_{2.5}$-bound PAHs, etc.. So we carried out a more detailed experiment with six sampling sites in the second year. In this paper, except for reporting the level and seasonal variations of OPEs at six sites, we paid more attention to investigate the relationships and correlations among the target compounds or with influence factors and illustrate the potential sources of OPEs in $PM_{2.5}$. For example, whether different functional areas affect the distributions of atmospheric OPEs, correlations of OPEs with environmental factors (temperature, wind, vapor pressure, boiling points, etc.), correlations of OPEs with $PM_{2.5}$ concentrations, correlations of

OPEs in $PM_{2.5}$ and soil, correlations of OPEs in indoor and outdoor air were all discussed. These differences are the innovation of this paper.

In the revised manuscript, the novelty has been added in lines 66-78: "Our previous study has investigated the OPEs concentrations in $PM_{2.5}$ at two sites (urban and suburban sites) in Chengdu (an economically fast growing city in southwest of China), and found that OPEs concentrations and profile were similar at two sites (Yin et al., 2015). But the influence factors and potential sources of OPEs in

$PM_{2.5}$ in Chengdu are still unclear. Therefore, in this study, $PM_{2.5}$ was collected over one year (October

2014 to September 2015) at six sites in Chengdu to: a) report the levels and composition profiles of

OPEs in urban air in the typical inland city; (b) obtain the seasonal and spatial variation of OPEs in

$PM_{2.5}$; (c) investigate the relationships and correlations among the target compounds or with influence factors; (d) illustrate the potential sources of OPEs in $PM_{2.5}$."

QA/QC: 1) As no surrogate standards were spiked prior to sample treatment, how did authors evaluate OPE recoveries from the analytical procedures?

**Response:** Lines 127-132, "Thorough QA/QC procedures for OPE analysis were conducted to ensure data quality. To evaluate the recovery efficiencies of analytical procedures, all samples were added with internal standard (TDCPP-$d_{15}$ and TPhP-$d_{15}$), and the accuracy was evaluated by their recoveries. The concentrations of the 7 OPEs were determined by an external standard method. The correlation coefficients of the standard curves of the seven OPE monomers were all greater than 0.990.

The recoveries of 7 OPEs and the internal standard were between 78.9% and 122.5%." was added.

Lines 135-137, "Field blanks were done at each site to evaluate the background contamination in the field. TBEP, TnBP and TEHP were detected in it. The level of them found in the blank were <15%

of the concentrations measured in all samples." was added.

There are only two internal standards, so we use them to ensure the recovery, but use external standard method to quantify the target compounds. In addition, a matrix blank was run in parallel with every batch of samples for the analysis of OPEs. Only TnBP was detected in the blanks, and the level of TnBP found in the blanks was <5% of the concentrations measured in all samples, which meant it was negligible. The correlation coefficients of the standard curves of the seven OPE profiles were all greater than 0.990. These all could ensure the accuracy of the data.

2) How was the matrix effect assessed and compensated?

**Response:** The matrix effect was assessed by the matrix blank experiment. The blank quartz membrane was added with the internal standard (TDCPP-d$_{15}$ and TPhP-d$_{15}$) and OPEs standard. After the whole pretreatment process, the recoveries of 7 OPEs and the internal standard were all between

70% and 120%. So the data was not corrected and the matrix effect was not compensated.

3) The data from field blanks were missing. PMF model: How were the uncertainties determined?

Which references were referred to for identification of sources associated with each factor? I also want to see the source profile of each factor.

**Response:** The field blanks were done which were prepared and installed in the same manner as the regular samples but without turning on the sampler motor. Due to the limited space of the paper, and the focus of this paper is not on the establishment of analytical methods, so it is simplified a lot in the QA/QC part. But we have done all the related experiments for QA/QC, and the results were good.

Lines 135-137, "Field blanks were done at each site to evaluate the background contamination in the field." TBEP, TnBP and TEHP were detected in it. The level of them found in the blank were

<15% of the concentrations measured in all samples." was added.

Lines 445-446, "The uncertainty is estimated by three methods: BS, disp and bs-disp" was added for PMF. The results are shown in the table below:

DISP results showed that the solution was stable because no swaps were present.

BS results showed that mapping over 80% of the factors indicated that the BS uncertainties could be interpreted and the number of factors may be appropriate.

All of the "Strong" species were selected for the BS-DISP error estimation. The number of DISP

and BS-DISP swaps was zero. BS-DISP highlight that the solution may be reliable due to there was no swaps across two factors.

Error estimation summary results

| BS-DISP Diagnostics: | | | | |
|---|---|---|---|---|
| # of Cases Accepted: | 100 | | | |
| % of Cases Accepted: | 100% | | | |
| Largest Decrease in Q: | -0.150999993 | | | |
| %dQ: | -0.067824623 | | | |
| # of Decreases in Q: | 0 | | | |
| # of Swaps in Best Fit: | 0 | | | |
| # of Swaps in DISP: | 0 | | | |
| Swaps by Factor: | 0 | 0 | | 0 |
| DISP Diagnostics: | | | | |
| Error Code: | 0 | | | |
| Largest Decrease in Q: | -0.005 | | | |
| %dQ: | -0.002245848 | | | |
| Swaps by Factor: | 0 | 0 | | 0 |
| BS Mapping: | | | | |
| | Factor 1 | Factor 2 | Factor 3 | Unmapped |
| Boot Factor 1 | 100 | 0 | 0 | 0 |
| Boot Factor 2 | 0 | 100 | 0 | 0 |
| Boot Factor 3 | 0 | 0 | 100 | 0 |

The source profile of each factor:

| Factor Profiles (% of species sum) from Base Run (Convergent Run) | | | | | | | |
|---|---|---|---|---|---|---|---|
| | TnBP | TCEP | TCPP | TDCPP | TPhP | TBEP | TEHP |
| Factor 1 | 28.69 | 70.95 | 70.72 | 31.01 | 58.34 | 0.00 | 70.93 |
| Factor 2 | 0.00 | 20.31 | 25.47 | 44.72 | 13.97 | 77.95 | 26.41 |
| Factor 3 | 71.31 | 8.73 | 3.81 | 24.27 | 27.69 | 22.05 | 2.66 |

Lines 450-455, the references were referred to for identification of sources associated with each factor. "Factor 1 was deduced to be the plastics/electrical industry and indoor source emissions (Esch,

2000; Stevens et al., 2006). Factor 2 contributed the most to TBEP (78.0%), followed by TDCPP

(44.7%), while it did not contribute to TnBP. Therefore, factor 2 was deduced as the food/cosmetics industry and traffic emissions (Marklund et al., 2005). Factor 3 contributes 71.7% of the total TnBP, which can be deduced as chemical industrial source (Regnery et al., 2011)."

Minor concern:

Line 8: "emerging contaminants"→"contaminant of emerging concern". OPEs have been produced for decades.

**Response:** We're sorry for the improper expression. Line 10, "emerging contaminants" has been revised to "OPEs are contaminants of emerging concern".

Line 9: "centers"→ "areas"

**Response:** Thanks for your advice. Line 11, "Centers" has been replaced by "areas".

Line 13... which TOGETHER made up..."

**Response:** Thanks for your advice. Line 15, "which made up" has been revised to "which together made up".

Line 18: OPEs can transfer from soil to air particles via suspension and volatilization as well.

Actually, authors mentioned this at Lines 303-304.

**Response:** Thanks for your advice. Line 20, "suggested the atmospheric $PM_{2.5}$ settlement is an important source of OPEs in soil" has been deleted. The sentence has been revised to "Very strong correlation ($R^2 = 0.98$, p<0.01) between the OPEs in soil and in $PM_{2.5}$ was observed."

Lines 32-35: A weird sentence, please rephrase it.

**Response:** Thanks for your advice. Lines 36-38, the sentence has been revised to "However, many scholars found that the residues of OPEs in the environment could cause toxic effects on organisms (WHO, 1991, 1998, 2000; Kanazawa et al., 2010; Van der Veen and de Boer, 2012; Du et al., 2015)".

Line 35: Reference is needed for the "OPE restrictions".

**Response:** Lines 40-41, three references were added "Some countries have legislated to restrict the usage of OPEs (Blum et al., 2019; Exponent, 2018; State of California, 2020)".

Lines 495-498, "Blum, A.; Behl, M.; Birnbaum, L. S.; Diamond, M. L.; Phillips, A.; Singla, V.;

Sipes, N. S.; Stapleton, H. M.; Venier, M. Organophosphate ester flame retardants: Are they a regrettable substitution for polybrominated diphenyl ethers? Environ. Sci. Technol. Lett. 2019, 6, 638-

649. " was added.

Lines 532-534, "Exponent. California bans flame retardants in certain consumer products. 2018,

Available      at:      https://www.exponent.com/knowledge/alerts/2018/09/california-bans-flame- retardants/?pageSize=NaN&pageNum=0&loadAllByPageSize=true (accessed February 15, 2020).

State of California. Safer consumer products (SCP) information management system. 2020.

Available at: https://calsafer.dtsc.ca.gov/cms/search/?type=Chemical (accessed February 21, 2020)."

were added.

Lines 38-39: Reference is needed.

**Response:** Thanks for your advice. Lines 44-47, references have been added: "The detection of

OPEs in Arctic and Antarctic snow samples and atmospheric particulate matter samples demonstrated that OPEs can be transported over long distances (Möller et al., 2012; Li et al., 2017). Studies on OPEs in oceans were carried out a lot, and the concentrations of particle-bound OPEs ranged from tens to thousands of ng m$^{-3}$ (Möller et al., 2011; 2012; Cristale J & Lacorte S., 2013; Li et al., 2017;

McDonough et al., 2018)".

Lines 583-592, "McDonough, C. A., De Silva, A. O., Sun, C., Cabrerizo, A., Adelman, D.,

Soltwedel, T., Bauerfeind, E., Muir, D. C. G., Lohmann, R.: Dissolved organophosphate esters and polybrominated diphenyl ethers in remote marine environments: Arctic surface water distributions and net transport through Fram Strait, Environ. Sci. Technol., 52, 6208-6216, https://doi.org/10.1021/acs.est.8b01127, 2018.

Möller, A., Sturm, R., Xie, Z., Cai, M., He, J., Ebinghaus, R.: Organophosphorus flame retardants and plasticizers in airborne particles over the northern pacific and Indian Ocean toward the polar regions:

evidence for global occurrence, Environ. Sci. Technol., 46, 3127-3134. 2012

Möller, A., Xie, Z., Caba, A., Sturm, R., Ebinghaus, R.: Organophosphorus flame retardants and plasticizers in the atmosphere of the North Sea, Environ. Pollut., 159, 3660-3665, https://doi.org/10.1016/j.envpol.2011.07.022, 2011." have been added.

Line 45: Which type of matrix is referred to for "Concentrations of OPEs in most cities..."I looked at the references cited, but not all of them talked about PM$_{2.5}$.

**Response:** This matrix is only for outdoor atmospheric environment.

Line 51, "Concentrations of OPEs in most cities..." has been revised to "Concentrations of atmospheric OPEs in most cities". Not all of the references we cited talked about PM$_{2.5}$, but they were all about OPEs in atmospheric particles.

Lines 54-56: How about "Chengdu is an important city in Southwest China due to its role as a national high-tech industrial base, a commercial logistics center, and a comprehensive transportation hub"?

**Response:** Thanks for your advice. Line 62, a website has been added:

"(https://en.wikipedia.org/wiki/Chengdu)."

Line82: Sampling intervals?

**Response:** Lines 94-100, "In each season, continuous sampling was carried out for about one week, except for rainy days. In autumn, the sampling duration was from October 23 to October 29, (no sample was obtained due to the rain on October 26 and 27); in winter, the sampling duration was from December 22 to December 30, 2014 (no sample was obtained due to the rain on October 25

and 26); in spring, the sampling duration was from March 25 to March 30, 2015; in summer, the sampling duration was from July 16 to July 24, 2015 (no sample was obtained due to the rain on July

21)" has been added. Each collection campaign lasted 23 h. The interval of each sample was 1h.

Line 86: Was the analytical method used here applied in any previous studies?

**Response:** Based on the references of Möller et al. (2012), we established the quantitative analysis method in the laboratory. The analytical method has been applied in our previous studies.

Li, S. P., Yin, H. L., YE, Z. X., Liang, J. F., Hao, Y. F.: GC-MS determination of 7 organic phosphate ester flame retardants in atmospheric particulates with chromatography purification: PTCA (Part B:

Chem Anal). 2015, 051(005), 581-585, https://doi.org/10.13227/j.hjkx.2015.10.003, 2015.

Yin, H. L., Li, S. P., Ye, Z. X., Yang, Y. C., Liang, J. F., You, J. J.: Pollution level and sources of organic phosphorus esters in airborne $PM_{2.5}$ in Chengdu City, Environ. Sci. (in chinese), 36, 3566-3572, https://doi.org/10.13227/j.hjkx.2015.10.003, 2015.

Möller, A., Sturm, R., Xie, Z., Cai, M., He, J., Ebinghaus, R.: Organophosphorus flame retardants and plasticizers in airborne particles over the northern pacific and Indian Ocean toward the polar regions:

evidence for global occurrence, Environ. Sci. Technol., 46, 3127-3134. 2012

Lines 93-94: How about "The latter eluate was collected and con-centrated by vacuumcondensing..."?

**Response:** The latter eluate was collected in a centrifugal tube and then concentrated to nearly dry by vacuum condensing equipment and then fixed volume to 200 μL with hexane. Then it was placed in a sample bottle to wait for the injection of gas chromatography-mass spectrometry (GC-MS).

Lines 111-113, it has been revised to "…the latter eluate (ethyl acetate/acetone) was collected.

The eluate was concentrated to nearly dry by vacuum-condensing equipment and then fixed volume to

μL with hexane for gas chromatography-mass spectrometry (GC-MS) (Shimadzu 2010plus, Japan)

analysis." in the manuscript.

Lines 114-118: Could concisely say "detected in virtually all the samples".

**Response:** Thanks for your advice. Lines 145-148, "Four OPEs (TCPP, TDCPP, TCEP and TnBP)

were detected in all samples (n=149), while TBEP was detected in all but one sample. Additionally,

TEHP was detected in 96.7% of samples overall and TPhP was detected in 98% of samples." has been revised to "Seven OPEs were detected in 96.7% - 100% of the samples (n=149)".

Lines 120-121: Did "The average value... four seasons mean "annual average level"?

**Response:** It means "seasonal average concentration", not "annual average level". Lines 151-153, it has been revised to "The seasonal average value of OPEs in $PM_{2.5}$ at each site was almost at the same level (5.8 $\pm$ 1.3 ng m$^{-3}$-6.9 $\pm$ 2.5 ng m$^{-3}$)". We have checked other places throughout the manuscript.

Line 141: Rephrase the first sentence.

**Response:** Line 176, it has been revised to "Non-chlorinated OPEs were the predominant OPEs across Chengdu city".

Lines 143-145: Explain the meaning of values in the parentheses.

**Response:** Line 179, it has been revised to "(annual media concentration: 2.3 ng m$^{-3}$, 35.3% of $\Sigma_7$

OPEs)".

Lines: 165-167: References?

**Response:** Line 202, two company websites for producing and selling OPEs have been added:

"https://show.guidechem.com/hainuowei, http://www.sinostandards.net/index.php".

Lines182-184: A recent study measuring an extended list of OPEs in the Great Lakes atmosphere also found that alkyl OPEs dominated OPE compositional profiles of urban air collected from Chicago and Cleveland (Wu et al. 2020; 10.1021/acs. est.9b07755).

**Response:** Thank you for your reminder. We have referred to the results of this study. For example, (1) lines 220-221, "Wu et al. (2020) also reported that alkyl OPEs dominated OPE

compositional profiles of urban air collected from Chicago and Cleveland." (2) Lines 247-252, "Wu et al. (2020) reported that median concentrations of $\Sigma$ OPEs for summer samples were up to 5 times greater than those for winter samples. The similar seasonal patterns were reported by Salamova et al.

(2014) for the atmospheric particle-phase OPE concentrations in samples collected from the Great

Lakes in 2012. A reasonable explanation is that OPEs are not chemically bound to the materials in which they are used and higher temperatures may facilitate their emission from buildings and vehicles." has been added. (3) Lines 290-293, "Interestingly, in this study, alkyl OPEs dominated both urban and suburban sites. This was extremely different from the results reported by Wu et al. (2020)

that alkyl OPEs dominated at urban sites, chlorinated OPEs were prevalent at rural sites, and aryl OPEs were most abundant at remote locations." has been added in the revised manuscript.

Line 208-210: OPE levels can be surely affected by temperature, so I suppose the authors would like to say "seasonal variations in OPE levels". Additionally, would meteorological parameters other than temperature result in the seasonal variations found in the present study?

**Response:** Thanks for your advice. Based on our experience, we also strongly agree that temperature and other meteorological factors will affect the level of pollutants in $PM_{2.5}$. However, the concentration of OPEs found in this study did not varied much in the four seasons, which was significantly different from other pollutants. Some literatures showed that the seasonal variations of

OPEs in some coastal cities were significantly affected by temperature (Liu et al., 2016; Wang et al.,

2019). For example, Wang et al. (2019) reported seasonality was discovered for OPEs in both gas phase and $PM_{2.5}$ with their concentrations higher in hot seasons in Dalian, which may due to the temperature-driven emission or gas-particle partitioning.

Lines 254-259, it has been revised to "In our study, the correlation analysis between the temperature, wind speed, wind direction and $\Sigma_7$OPEs concentrations has been done. The results showed statistically significant negative correlations between temperature and $\Sigma_7$ OPEs (R= -0.355, p<0.01).

The lowest concentrations of $\Sigma_7$OPEs and individual compound were observed in summer suggesting the OPEs level was not driven by the temperature-driven emission. Gas-particle partitioning and local emission sources may contribute to the variation." In addition, other meteorological parameters with high contributions to the seasonal variations were not found in the present study.

Lines 236-238: Has been mentioned before. Lines 238-248: Out of place here. Could be moved to section 3.1.

**Response:** Thanks for your advice. Lines 295-297 have been deleted.

Line 257: Need reference to support "they tend to be adsorbed in $PM_{2.5}$".

**Response:** Thanks for your advice. Line 316, reference has been added: " (Wang et al., 2019)".

Lines 675-677, "Wang, Y., Bao. M. J., Tan. F., Qu. Z. P., Zhang. Y. W., Chen. J. W.: Distribution of organophosphate esters between the gas phase and $PM_{2.5}$ in urban Dalian, China, Environ. Pollut., https://doi.org/10.1016/j.envpol.2019.113882, 2019."

Line 315: Other factors may lead to such difference between indoor and outdoor OPEs. For example, TBEP has the shortest atmospheric half-lives, which may explain why its dominance in indoor samples was not observed for the outdoor counterparts.

**Response:** Of course, other factors may also cause differences in the content of indoor and outdoor OPEs. Lines 385-395 have been revised to "Except for the different usage of OPEs, many factors may also lead to differences between indoor and outdoor OPEs. For example, TBEP has the shortest atmospheric half-lives, which may explain why its dominance in indoor samples was not observed for the outdoor counterparts. Studies in Swedish (Wong et al., 2018) reported the concentrations of OPEs in indoor air were TCPP > TCEP > TBEP > TnBP> TPhP, and in outdoor urban air were TBEP > TCPP > TCEP > TnBP > TPhP which also indicated the differences of OPEs profile in indoor and outdoor air. They found that activities in the building, e.g. floor cleaning, polishing, construction, introduction of new electronics and changes in ventilation rate could be key factors in controlling the concentration of indoor air pollutants, while the observed seasonality for OPEs in outdoor air was due to changes in primary emission."

Lines 350-356: References are required for identification of possible sources associated with each factor.

**Response:** Thanks for your advice. Lines 433-440, references have been added: "Factor 1 can represent the sources of OPEs from the plastic industry, interior decoration and traffic emission, with the contribution ratio of 34.5% (Marklund et al., 2005; Regnery et al., 2011; CEFIC, 2002). Factor 2 has higher load on TnBP, TEHP and TPhP. The highest load was on TnBP, which is often used as a high-carbon alcohol defoamer, mostly in industries that do not come in contact with food and cosmetics, as well as in antistatic agents and extractants of rare earth elements. TEHP can be used as an antifoaming agent, hydraulic fluid and so on. TPhP is typically used in electrical and electronic products, or plastic film and rubber (Esch, 2000; Stevens et al., 2006; Wei et al., 2015)."

---

## Referee Report (RR1)

**Reviews:** The revised manuscript has addressed most of the concerns and questions from previous reviewers except for the writing part. The scientific writing (grammar, choice of wording, etc.) needs to be greatly improved. Some of the sections, e.g.QA/QC and discussion is a little weak.

**Specific comments on the manuscript**

1. Title, what do you mean by seasonal of OP esters? Maybe better change the title to "Seasonal distributions and …."?
2. Abstract, line 14, significant numbers for the measured concentrations should be consistent.
3. Abstract, line 18, "high levels" instead of "high level".
4. Abstract, line 19, delete "Very".
5. Abstract, line 25-27, please check grammar and the selection of wording. E.g. line 25, maybe use "from the inland city" instead of "in the inland city". Line 26, delete "the". Line 27, change "whose".
6. Intro, line 41, rephrase line 41.
7. Intro, line 44, change "appeared".
8. Intro, line 47-line 49, those three sentences should be linked with some connection words, at least not by commas.
9. Intro, line 51, change "especially" to "mainly".
10. Intro, line 51-52, change "little attention was paid to."
11. Intro, line 53, you may delete "typical".
12. Intro, line 56, "the important" change to "an important".
13. Intro, line 64, "profiles"?
14. Materials and Methods. Chemicals. Line 72-79, rewrite this paragraph.
15. Materials and Methods. Sample Collection, line 92-94, please rephrase to be more concise. And what do you mean by "weather conditions could represent typical weather conditions of the season"?
16. Materials and Methods. QA/QC, delete line 117.
17. Materials and Methods. QA/QC, line 118, recovery efficiency? Recovery=extraction efficiency. "Add" might be better changed to "Spiked".
18. Results and Discussion, line 137, change "in the air of Chengdu city".
19. Results and Discussion, line 151-161, did those studies you cited serve as support for the sentence in line 145-147? Could you provide more discussion instead of listing the literature values?
20. Results and Discussion, line 190, please rephrase.

---

## Author Response (AR2)

Editor Decision: Reconsider after major revisions (03 Sep 2020) by Jason Surratt

Comments to the Author:

Dear Authors,

The reviewers mostly feel you have addressed the comments initially raised on your manuscript. However, one reviewer noted that the English grammar was very poor in many sections of the manuscript. I concur with this assessment and feel further English grammar editing is needed before full publication can be considered. I kindly ask the authors to either reach out to a native-English speaking colleague to edit your manuscript or use a professional service. The number of grammar writing errors make the article difficult to read. Please let me know if you have any questions. One of the reviewers noted many places where the English writing needs to be fixed, but I should note to the authors it is not the duty of the reviewers to help with rewriting of the manuscript.

Most sincerely, Jason Surratt

**Response**

Dear Surratt,

Thanks very much for your kindness and your valuable advice. Now this paper has been edited for grammar, phrasing, and punctuation by American Journal Experts. In addition, many edits were made to further improve the flow and readability of the text. I very much hope that the manuscript can meet the requirements of your journal so that it can continue to be reviewed and published. Thanks again.

The proof is as following:

[Figure]

September 16, 2020

Dear Hongling Yin,

Thank you for choosing American Journal Experts. This manuscript, titled "Measurement report: Seasonality, distribution and sources of organophosphate esters in PM2.5 from an inland urban city in Southwest China," is very interesting. The paper was edited for grammar, phrasing, and punctuation. In addition, many edits were made to further improve the flow and readability of the text. Below, we highlight the areas of this paper that we focused on in our edit.

In cases where the meaning of the text was not clear, revisions were made to convey the information with increased clarity and reduced ambiguity.

Articles are an important aspect of the English language, including the definite article "the" and the indefinite articles "a" and "an." Our edits focused on improving article use, which is often strongly dependent on context and field conventions.

Some edits were made to improve conciseness by trimming unnecessary words and streamlining the flow of your manuscript.

Comments were left if further clarification would be helpful or confirmation of the meaning of the text was necessary. Please review these comments and all our changes carefully for more detailed suggestions, as well as to ensure that the final version of the manuscript is fully accurate.

Thank you again for using our editing services; we wish you the best of luck with your submission.

Best regards,

Katie D.
Senior Editor
American Journal Experts

Best Regards.

Yours,

Sincerely,

Hongling Yin

2020-9-22

**Measurement report: Seasonality, distribution and sources of organophosphate esters in PM$_{2.5}$ from an inland urban city in Southwest China**

Hongling Yin, Jiangfeng Liang, Di Wu, Shiping Li, Yi Luo, Xu Deng

College of Resources and Environment, Chengdu University of Information Technology, Chengdu, Sichuan, 610025, China

*Correspondence: Hongling Yin (yhl@cuit.edu.cn)*

**Abstract.** Organophosphate esters (OPEs) are contaminants of emerging concern, and studies have concluded that urban areas are a significant source of OPEs. Samples were collected from six ground-based sites located in Chengdu, a typical rapidly developing metropolitan area in Southwest China, and

were analysed for seven OPEs in atmospheric PM$_{2.5}$ ($\Sigma_7$ OPEs). The concentrations of $\Sigma_7$ OPEs in PM$_{2.5}$ ranged from 5.83 to 6.91 ng m$^{-3}$, with a mean of 6.6 ± 3.3 ng m$^{-3}$, and the primary pollutants were tris-(2-butoxyethyl) phosphate (TBEP), tri-n-butyl phosphate (TnBP), tris-(2-chloroethyl) phosphate (TCEP) and tris-(2-chloroisopropyl) phosphate (TCPP), which together made up more than 80% of the $\Sigma_7$ OPEs. The concentrations of $\Sigma_7$ OPEs were higher in autumn/winter than in summer. Nonparametric tests showed that there was no significant difference in $\Sigma_7$ OPE concentrations among the six sampling sites, but the occurrence of unexpectedly high levels of individual OPEs at different sites in autumn might indicate noteworthy emissions. A very strong correlation (R$^2$ = 0.98, $p$<0.01) between the OPEs in soil and in PM$_{2.5}$ was observed. Backward trajectory analysis indicated that the OPEs in PM$_{2.5}$ were mainly affected by local sources. Principal component analysis (PCA) revealed that the OPEs in PM$_{2.5}$ were largely sourced from the plastic industry/interior decoration/traffic emission (34.5%) and the chemical, mechanical and electrical industries (27.8%), while the positive matrix factorization (PMF) model revealed that the main sources were the plastics industry/indoor source emissions, the food/cosmetics industry, and industrial emissions. In contrast to coastal cities, sustained and stable high local emissions in the studied inland city were identified, which is particularly noteworthy. Chlorinated phosphates, especially TCPP and TCEP, had a high content, and their usage and source emissions should be controlled.

**1. Introduction**

With the prohibition of brominated flame retardants, the production of and demand for organophosphate esters (OPEs) have rapidly increased in recent years (Wang et al., 2012). OPEs are widely distributed in the environment and have been detected in air (Guo et al., 2016; Li et al., 2017), water (Wang et al., 2013; Li et al., 2014), soil (Yin et al., 2016), sediment (Cristale J. et al., 2013; Celano R, et al., 2014) and organisms (Kim et al., 2011). However, many scholars have found that OPE residues in the environment can cause toxic effects on organisms (WHO, 1991, 1998, 2000; Kanazawa et al., 2010; Van der Veen and de Boer, 2012; Du et al., 2015). Some countries have enacted legislature to restrict the usage of OPEs (Blum et al., 2019; Exponent, 2018; State of California, 2020). Nevertheless, the production and usage of OPEs in China are still on the rise.

As OPEs are synthetic substances, the only source of OPEs in the environment is anthropogenic emissions. The detection of OPEs in Arctic and Antarctic snow samples and atmospheric particulate matter samples demonstrated that OPEs can be transported over long distances (Möller et al., 2012; Li et al., 2017). Many studies on OPEs in oceans have been carried out, and the concentrations of particle-bound OPEs range from tens to thousands of ng m$^{-3}$ (Möller et al., 2011; 2012; Cristale & Lacorte, 2013; Li et al., 2017; McDonough et al., 2018). Some researchers noted that high concentrations of OPEs (thousands of ng m$^{-3}$) originated from air flow from the mainland (Möller et al., 2012; Lai et al., 2015). In addition, studies have proven that urban areas have the highest OPE pollution. However, until now, only a few papers have reported the concentration and distribution of OPEs in urban atmospheric PM$_{2.5}$. Concentrations of atmospheric OPEs in most cities were lower than 10 ng m$^{-3}$; a higher concentration of 19.2 ng m$^{-3}$ was observed at a suburban site in Shanghai, and a concentration of 49.1 ng m$^{-3}$ was observed in Hong Kong (Ohura et al., 2006; Salamova et al., 2014b; Marklund et al., 2005; Shoeib et al., 2014; Yin et al., 2015; Liu et al., 2016; Ren et al., 2016; Guo et al., 2016; Wong et al., 2018). To date, most studies in China have focused on OPEs in the Yangtze River Delta and Pearl River Delta, especially eastern coastal cities, while little attention has been paid to western inland cities.

Chengdu is a typical inland city located in Southwest China. This megacity is the capital of Sichuan Province, covers an area of 14,335 square kilometres and has a permanent population of 16.33 million. As an important national high-tech industrial base, commercial logistics centre and comprehensive transportation hub designated by the State Council, Chengdu is the most important central city in the western region (https://en.wikipedia.org/wiki/Chengdu). Liu et al. (2016) investigated three chlorinated OPEs in the atmosphere at 10 urban sites in China during 2013 – 2014 and observed the highest annual mean concentrations in Chengdu (1,300 ± 2,800 ng m$^{-3}$). However, there is still a lack of information regarding the levels, sources, and fate of OPEs in Southwest China, which may obviously differ from those of coastal cities or ocean locations. Our previous study investigated OPE concentrations in PM$_{2.5}$ at two sites (urban and suburban sites) in Chengdu (a city experiencing fast economic growth in Southwest China) and found that the OPE concentrations and profiles were similar at the two sites (Yin et al., 2015). However, the influencing factors and potential sources of OPEs in PM$_{2.5}$ in Chengdu are still unclear. Therefore, in this study, PM$_{2.5}$ was collected over one year (October 2014 to September 2015) at six sites in Chengdu to a) report the levels and composition profiles of OPEs in urban air in a typical inland city; (b) obtain the seasonal and spatial variations in OPEs in PM$_{2.5}$; (c) investigate the

[revised manuscript text omitted]

in the summer in Guangzhou, and Javier et al. (2018) found that the OPEs in spring generally exhibited the lowest concentrations in Bizerte, Tunisia, probably linked to the influence of local meteorological conditions and, to a lesser extent, air mass trajectories.

Although the Kruskal-Wallis test showed no significant variation in $\Sigma_7$ OPE concentrations across the city, spatial differences were identified in this study. For example, TnBP and TCPP had significant differences among the six sites. In addition, higher concentrations and more dispersed patterns of most OPEs were observed in autumn and winter than in summer (Fig. 3). The concentrations of TEHP in autumn at the eastern and northern sampling sites were more dispersed than those at other sites. The same dispersion pattern was observed for TBEP in winter at the western sampling site, TPhP in autumn at the suburban sampling site, and TnBP in autumn at the eastern sampling site, suggesting that extra emission sources existed in autumn or winter. Considering the layout of Chengdu, which spreads out from the central area along the loop line (the first ring road, the second ring road and the third ring road), the uniform patterns of OPE levels and distribution across the city are understandable. Different types of industrial parks in different directions in Chengdu may be the reason for the spatial differences in OPEs. For example, in eastern Chengdu, there are automobile industrial parks and other large industrial parks, while logistics and shoemaking industrial parks are located in the suburbs. The occurrence of unexpectedly high levels of individual OPEs at different sites in autumn might indicate noteworthy emissions. The spatial and seasonal variations in individual OPEs suggest that OPE control and management measures should be taken. Interestingly, in this study, alkyl OPEs dominated at both urban and suburban sites. This finding was extremely different from the results reported by Wu et al. (2020), in which alkyl OPEs dominated at urban sites, chlorinated OPEs were prevalent at rural sites, and aryl OPEs were most abundant at remote locations.

Many studies have focused on halogenated OPEs due to their persistence, bioaccumulation, and potential human health effects, and they dominate the OPE profile in the air of many cities and other areas (Li et al., 2017). Liu et al. (2016) reported that the sum of the concentrations of three halogenated OPEs at 10 urban sites ranged from 0.05 to 12 ng m$^{-3}$, suggesting that the highest production volume and widest application of OPEs have led to large emissions of OPEs in China in recent years. However, in our study, the mean concentrations of halogenated, alkylated and aryl OPEs were 2.4 ± 1.4 ng m$^{-3}$, 3.7 ± 2.1 ng m$^{-3}$, and 0.5 ± 0.4 ng m$^{-3}$, respectively, which showed that alkylated OPEs dominated the profile of OPEs in PM$_{2.5}$ in Chengdu. The most notable seasonal variation was observed for alkyl phosphates, followed

Though

批注 [Ed2]: Please note that the font changes here. Please consider using a consistent font type and size throughout the manuscript.

[revised manuscript text omitted]